# K-HALU: Multiple Answer Korean Hallucination Benchmark for Large Language Models

**Jaehyung Seo**
Korea University
Seoul, Republic of Korea
seojae777@korea.ac.kr

**Heuiseok Lim**[*]
Korea University
Seoul, Republic of Korea
limhseok@korea.ac.kr

## Abstract

Recent researchers and companies have been developing large language models (LLMs) specifically designed for particular purposes and have achieved significant advancements in various natural language processing tasks. However, LLMs are still prone to generating hallucinations—results that are unfaithful or inconsistent with the given input. As a result, the need for datasets to evaluate and demonstrate the hallucination detection capabilities of LLMs is increasingly recognized. Nonetheless, the Korean NLP community lacks publicly available benchmark datasets demonstrating the faithfulness of knowledge-based information. Furthermore, the few existing datasets that evaluate hallucination are limited in their access to the entire dataset, restricting detailed analysis beyond simple scoring, and are based on translated English knowledge. To address these challenges, we introduce **K-HALU**[1], a Korean benchmark designed to evaluate LLMs' hallucination detection in Korean. This benchmark contains seven domains, considering the faithfulness of statements based on knowledge documents compiled from Korean news, magazines, and books. For more strict evaluation, 40% of the dataset is structured as multiple-answer questions, requiring models to select all possible correct answers from the given options. Our empirical results show that open-source LLMs still struggle with hallucination detection in Korean knowledge, emphasizing the need for a more detailed analysis of their limitations.

## 1 Introduction

Large language models (LLMs) have achieved remarkable advances, surpassing human capabilities in various natural language processing (NLP) tasks (Ouyang et al., 2022; OpenAI, 2023). Recently, compact and high-performing open-source LLMs have emerged (Touvron et al., 2023; Jiang et al., 2023), and researchers and companies are leveraging these models to develop their own purpose-specific systems (Kim et al., 2024; Research et al., 2024). Within the Korean NLP community, over 2,000 models were evaluated on the Open Ko-LLM leaderboard (Park et al., 2024) from October 2023 to August 2024, illustrating an increase in proprietary LLM development.

However, as a result of limited parameter sizes and constrained training data, open-source LLMs continue to struggle with the problem of generating hallucinated outputs (Ji et al., 2023a; Zhang et al., 2023; Li et al., 2023). Hallucinated outputs cannot guarantee faithfulness with the provided data and often include unsupported or unverifiable content (Huang et al., 2023a). Hallucinations present a significant threat to the reliability and practical applications of LLMs (Chen et al., 2024), and LLMs with relatively fewer parameters or incomplete data are even more vulnerable to this phenomenon (Rawte et al., 2023; Guerreiro et al., 2023).

A more pressing issue in the Korean NLP community is the lack of benchmarks to verify the potential risks associated with hallucinations in these numerous proprietary LLMs. The few available Korean datasets related to hallucinations are typically closed and used solely for leaderboard-style scoring, limiting access to the data for detailed analysis (Park et al., 2024). Furthermore, most benchmarks focus on parametric knowledge and linguistic nuances specific to the English-speaking world,

---

[*]Corresponding Author
[1]https://github.com/J-Seo/K-HALU

Table 1: Examples of K-HALU benchmark according to the instruction type for selecting hallucinated statements and faithful statements. This table has been translated from Korean into English for the convenience of non-Korean speakers. (Refer to the Korean version in Appendix G).

| |
|---|
| **### Hallucinated statements selection type — Society Domain ###** |
| **#date**: 20210711 |
| **#document**: The Korean Association for Public Administration was established in 1956 ... Red tape is a symbol of bureaucratic formalism. |
| **#instruction**: Select the hallucinated statements that differ from or are unsupported in relation to the content of the given document. Note that there can be multiple hallucinated statements. |
| **#1**: Professor Park Soon-ae is working to expand women's participation in the public sector. |
| **#2**: Professor Park Soon-ae went to the United States to pursue a Ph.D. while raising two children. |
| **#3**: Professor Park Soon-ae declares an intention to reform the bureaucratic field in the 3G era. |
| **#4**: Professor Park Soon-ae points out that administrative convenience is an issue in Korea. |
| **#5**: Professor Park Soon-ae states that civil servants prefer maintaining regulations over deregulation. |
| **#label**: [1,2] (Index number of choice) / **#choice_number**: [2,3] (Numbering of choice) |
| **### Faithful statements selection type — International Domain ###** |
| **#date**: 20150819 |
| **#document**: Google is launching a new 'Android One' smartphone in six African countries ... It is one of the major projects being pursued. |
| **#instruction**: Select the faithful statements that correspond to the information identifiable from the given document. Note that there may be multiple correct statements. |
| **#1**: Google is selling the 'Hot 2' model in six North American countries, including the Canada and Mexico. |
| **#2**: The price of the newly launched model is under 100 dollars. |
| **#3**: Google is launching a new 'iPhone One' smartphone in six African countries. |
| **#4**: The 'Hot 2' model, produced by Infinix, features a 10-inch touchscreen and a 5GHz quad-core processor. |
| **#5**: A satellite internet service project is underway in Kampala, the capital of Uganda. |
| **#label**: [2] (Index number of choice) / **#choice_number**: [3] (Numbering of choice) |

making them less ideal as resources for evaluating underrepresented languages (Etxaniz et al., 2024). In particular, English hallucination benchmark datasets are challenging to apply to the Korean language caused by linguistic and socio-cultural differences (Hendrycks et al., 2021; Seo et al., 2024). The absence of publicly available Korean hallucination benchmarks restricts the ability to evaluate the reliability of LLMs thoroughly, hinders the continuous accumulation of findings needed for improvement, and makes it challenging to capture the robustness of LLMs in detecting hallucinations.

To overcome these limitations, we introduce the multiple-answer Korean hallucination benchmark for large language models (**K-HALU**). K-HALU consists of 2,170 test samples, each including a textual document, a publish date, an instruction, and statements. The textual documents are sourced from seven knowledge domains—Culture, Economy, History, International, Medical, Society, and Technology—from Korean news, magazines, and books. As described in Table 1, LLMs should select the appropriate statements in a multiple-choice format, considering the given textual document and publish date. Unique to K-HALU is the multiple-answer question setup, which requires LLMs to identify all possible correct answers. This approach ensures a more rigorous evaluation of model reliability and assesses the models' ability to recognize multiple simultaneous hallucinations. Also, K-HALU includes statements considering the publish date, allowing us to determine whether LLMs maintain temporal consistency in faithfulness.

We evaluate open-source multilingual LLMs frequently used in the Korean NLP community, such as Llama2 (Touvron et al., 2023), Llama3 (AI@Meta, 2024), Mistral (Jiang et al., 2023), and Korean-centric models such as KULLM3 (Kim et al., 2024) and ExaOne (Research et al., 2024). Additionally, we test closed API models with high performance and usability, including GPT-3.5 Turbo (Ouyang et al., 2022), GPT-4 Turbo, and GPT-4 omni (OpenAI, 2023). The results reveal that open-source LLMs exhibit low accuracy, with less than 35% in our evaluations, and perform particularly poorly—under 15%—on instruction types designed to differentiate hallucinated statements. Compared to API models such as GPT-3.5 and beyond, open-source models present a performance gap exceeding 27%, showing greater weakness to hallucination as the number of answers increases.

## 2 RELATED WORK

**Hallucination in NLP**    With the prominent advancements of LLMs and their applicability to various tasks, the importance of research on hallucinations in NLP has increased significantly. Maynez

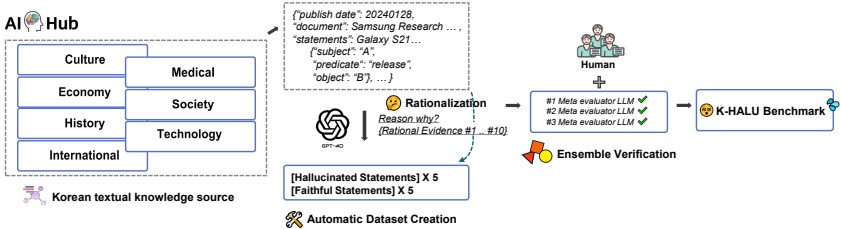

Figure 1: Overview of K-HALU benchmark construction pipeline.

et al. (2020) highlighted issues of faithfulness and factuality in abstractive summarization, while Raunak et al. (2021) pointed out similar concerns in neural machine translation, emphasizing that language model-based natural language generation can lead to hallucinations. Since then, research focusing on hallucinations in natural language generation tasks has become more prominent (Zhao et al., 2020; Shuster et al., 2021; Liu et al., 2021; Fabbri et al., 2022; Zhang et al., 2022). Following the emergence of outstanding generative LLMs (Ouyang et al., 2022), the definition and scope of hallucinations in NLP have been discussed in greater detail (Ji et al., 2023a; Huang et al., 2023a; Zhang et al., 2023). Ji et al. (2023a) categorized hallucinated outputs into two types: intrinsic hallucinations, which result from conflicts with the source content, and extrinsic hallucinations, which include information that cannot be verified by the source content. They focus on the contributors to hallucinations and explore task-centric mitigation strategies. Huang et al. (2023a) explored the causes and solutions to hallucinations based on faithfulness and factuality, while Zhang et al. (2023) classified hallucinations arising from conflicts between input, context, and factual knowledge with the model's generated output. Subsequent studies have actively explored hallucination detection (Huang et al., 2023b; Manakul et al., 2023; Jiang et al., 2024; Chen et al., 2024) and mitigation strategies (Maheshwari et al., 2023; Chuang et al., 2023; Ji et al., 2023b; Choubey et al., 2023).

Several English hallucination benchmarks have been developed to evaluate language models in various tasks. These include Fever (Thorne et al., 2018), which evaluates factual consistency against textual sources, QAGS (Wang et al., 2020), which measures factual inconsistencies in summaries, SummEval (Fabbri et al., 2021), which assesses the quality of summaries between human evaluators and models, FaithDial (Dziri et al., 2022), a dialogue-focused dataset for evaluating response faithfulness, HaluEval (Li et al., 2023), which determines faithful hallucination presence in model-generated samples, and FELM (Zhao et al., 2024), which measures hallucination across multiple domains. However, research that provides specific insights into hallucinations in the Korean language is still lacking, and there is a significant shortage of publicly available benchmark datasets that could serve as the foundation for hallucination studies in Korean.

**Korean Benchmark** Benchmarks serve as essential tools for the quantitative assessment of the strengths and weaknesses of LLMs, offering critical insights into the future direction of NLP research and development (Zellers et al., 2019; Son et al., 2024a). Existing Korean benchmarks have evaluated models based on linguistic skills or language understanding (Park et al., 2022), and tasks involving universal reasoning were often translated for applicability to other languages (Conneau et al., 2018; Ham et al., 2020; Ponti et al., 2020; Seo et al., 2022; Park et al., 2024). However, the need for native knowledge derived from textual sources written in Korean, beyond simple machine translation, has increasingly been recognized within the Korean NLP community for more precise performance measurement and knowledge verification (Son et al., 2024a; Seo et al., 2024). Tasks in benchmarks such as KorNLI & KorSTS (Ham et al., 2020), Korean-CommonGEN (Seo et al., 2022), and Ko-H5 (Park et al., 2024) are based on reasoning and have been constructed by translating existing datasets. KoBBQ (Jin et al., 2024) addresses biases using partial translations specific to the Korean cultural context. Meanwhile, benchmarks such as HAE-RAE (Son et al., 2024b), KMMLU (Son et al., 2024a), and KoCommonGEN v2 (Seo et al., 2024) have been developed using Korean textual sources and human annotators. However, these benchmarks do not evaluate hallucinations in Korean. Ko-TruthfulQA (Lin et al., 2022; Park et al., 2024), which partially addresses elements of hallucination, is a closed dataset, and even the latest Open Ko-LLM leaderboard, which could verify reliability, does not provide access to the datasets. Thus, we propose a new Korean hallucination benchmark **K-HALU** and plan to release the dataset and evaluation code to the public entirely.

Table 2: Number of K-HALU Benchmark examples according to knowledge domain, instruction type, and multiple-answer questions.

| K-HALU Benchmark | Culture | Economy | History | International | Medical | Society | Technology | Total |
|---|---|---|---|---|---|---|---|---|
| - # examples | 300 | 299 | 300 | 329 | 325 | 317 | 300 | 2,170 |
| **Instruction type** | | | | | | | | |
| - # hallucination select | 150 | 150 | 150 | 165 | 162 | 158 | 150 | 1,085 |
| - # fact select | 150 | 149 | 150 | 164 | 163 | 159 | 150 | 1,085 |
| **Multiple-answers** | | | | | | | | |
| - # single-answer | 180 | 179 | 180 | 197 | 196 | 190 | 180 | 1,302 |
| - # two-answers | 90 | 90 | 90 | 100 | 98 | 95 | 90 | 653 |
| - # three-answers | 30 | 30 | 30 | 32 | 31 | 32 | 30 | 215 |

# 3    K-HALU BENCHMARK

**K-HALU** is a hallucination detection benchmark composed of 2,170 multiple-choice tasks, where there can be more than one correct answer. As shown in Table 2, the seven domains are sourced from Korean textual knowledge and include Culture, Economy, History, International, Medical, Society, and Technology. The Culture domain covers topics related to entertainment, literature, and films; the Economy addresses issues such as administration, corporations, real estate, and the market; History contains Korean, world, and East Asian history; International deals with diplomacy, global corporations, and foreign affairs; Medical focuses on medical knowledge, pharmaceutical research, and health tips; Society covers education, North Korea, self-development, and interest conflicts; and Technology includes patents, research papers, and IT services as the textual sources. Each domain contains an average of 310 examples, and each example consists of the following components: "id," "document," "date," "instruction," "five statements," "choice number", and "label". All documents include the publish date. The statements are categorized as either hallucinated or faithful, and the proportion of these categories varies depending on the instruction type and the number of correct answers. Details regarding the policies and responsibilities related to the use of the K-HALU dataset are provided in §5 Ethics Statements.

## 3.1    TASK DEFINITION

The objective of the K-HALU task is for LLMs to use the provided document and publish date as source knowledge to discriminate the faithfulness of the given statements and detect hallucinations. K-HALU employs a multiple-answer question evaluation format, allowing for more than one correct answer. LLMs are required to select all possible correct statements based on the given instruction's question type. Given that the instruction demands "all possible correct answers," even a single incorrect selection by the LLM will result in an incorrect response, reflecting the strict nature of this evaluation. For example, if a question has three correct answers, selecting only two correctly will be classified as incorrect, as all correct answers must be identified for the response to be considered correct (the order of the selected answers is not taken into account during scoring).

## 3.2    DATASET CREATION

**Source Dataset**   We utilized the publicly available Knowledge Graph-to-Text dataset from AI-Hub, an integrated AI platform operated by the Korean National Information Society Agency (NIA), as our textual source[2]. This dataset consists of textual documents, including news articles, magazine articles, and books, describing faithful relationships in the form of statements. Each statement is tagged as subject, predicate, and object, forming a triples-context pair. Each document contains one or more triples-context pairs, amounting to a total of 300,178 samples, each of which has processed human labeling and data refinement. Among these samples, 89.8% are news articles, 4.9% are magazine articles, and 5.3% are sourced from books. The textual documents are divided into seven knowledge domains: Culture, Economy, History, International, Medical, Society, and Technology, with all domains—except for History—containing over 40,000 samples. Each sample includes one or more human-annotated statements that describe the corresponding textual document, and each

---

[2]https://www.aihub.or.kr/aihubdata/data/view.do?currMenu=115&topMenu=100&aihubDataSe=data&dataSetSn=71728

statement is labeled with tags for subject, object, and predicate. Out of 300,178 samples, we used 67,740 training samples to create options, considering domain proportions.

To construct the five answer options, we extracted documents from each domain (except for History) that contain at least five human-annotated statements. Since the History domain had a relatively smaller number of source data samples, we extracted documents that included at least two human-annotated statements. From each domain, we selected an average of 310 documents as the final textual sources, excluding those with statements that referenced content not mentioned in the text.

**Hallucinated Statement**   To create hallucinated statements that violate faithfulness, we leveraged the human-annotated statements, tagging labels, and published dates from the extracted documents as a basis for modification. Using GPT-4 omni (`gpt-4o-2024-05-13`) (OpenAI, 2023), we transformed the human-annotated statements into factually incorrect or unfaithful statements, misrepresenting event dates based on the publish date, or including unverifiable content. We generated five hallucinated statements per document. For the History domain, we added hallucinated statements that were directly generated without referencing human-annotated statements.

**Faithful Statement**   To prevent overfitting or unintended cheating from model training on the original dataset, we avoid using the human-annotated statements literally as faithful statements. Instead, we employed GPT-4 omni to either modify the existing human-annotated statements or generate faithful statements with high inferential quality that did not overlap with the originals. This process involved expressing tags at a higher conceptual level or paraphrasing while maintaining the same context. The faithful statements generated included higher-level reasoning, such as inferring event dates based on the publication date or combining information from multiple sentences. As with the hallucinated statements, we generated five faithful statements per document.

**Instruction and Multiple Answers**   We created instruction types that evenly required selecting hallucinated or faithful statements for each domain. We adjusted the number of hallucinated and faithful statements for each sample, considering the proportion of multiple-answer questions. These proportions were based on MultiSpanQA (Li et al., 2022), where the distribution of answer spans is 58%/35%/7%, and CLEAN (Luo et al., 2024), which uses 46% of question-answer pairs as multi-answer instances. We set the distribution of the multiple-answer question with one correct answer at 60%, two correct answers at 30%, and three correct answers at 10%, ensuring that more than two answers comprised 40% of the dataset. For example, if the $i^{th}$ test example's instruction requires selecting faithful statements and the label has two correct answers, two of the five statements are faithful, and three are hallucinations. To minimize bias from differences in token counts between options, we selected the final candidate options by choosing statements closest to the average length of the ten generated statements.

The final dataset comprises 2,170 examples constructed through the aforementioned process. The instruction types are divided into two categories based on the type of statements required, and the distribution of correct answers (1, 2, or 3) follows a 6:3:1 ratio with varying proportions of hallucinated and faithful statements across the five options.

### 3.3   QUALITY CONTROL

We implemented two mechanisms within the dataset generation pipeline to enhance the quality of automatic creation: (1) rationalization and (2) ensemble verification.

**Rationalization**   As illustrated in Figure 1, GPT-4 omni was instructed to provide the evidence for each statement generated by the generation of hallucinated and faithful statements. This approach aims to enhance the model's reasoning capabilities during the automatic execution of statements, ensuring that it consistently produces high-quality outputs by utilizing the rationale it established as contextual knowledge (Lei et al., 2016; Wang et al., 2023; Schimanski et al., 2024). Moreover, the generated rationale serves as a valuable resource for assisting both the meta-evaluator and human annotator in verifying the correctness of the outputs during the ensemble verification process.

**Ensemble Verification**   To further improve the quality of the final dataset samples, we conducted cross-validation by ensemble three top-performing models as meta-evaluators (Dutschmann

Table 3: Ensemble verification results for quality control across domains in the final dataset. The quality acceptance rate represents the proportion of examples in the dataset that received a verification score of 1. Underline indicates the lowest quality scores

| Quality acceptance rate | Culture | Economy | History | International | Medical | Society | Technology | Average |
|---|---|---|---|---|---|---|---|---|
| GPT-4 (gpt-4-06-13) | 99.54% | 99.59% | 98.94% | 99.54% | 99.35% | 99.08% | 99.72% | 99.39% |
| ChatGPT-4 (chatgpt-4o-latest) | 99.54% | 99.63% | 99.31% | 99.49% | 99.03% | 99.08% | 99.45% | 99.36% |
| GPT-4 omni (gpt-4o-2024-08-06) | 99.45% | 99.45% | 99.22% | 99.54% | 99.22% | 98.99% | 99.59% | 99.35% |

et al., 2023; Manakul et al., 2023; Gilardi et al., 2023). The selected models included GPT-4 (gpt-4-06-13), which demonstrates state-of-the-art performance in both benchmark tasks and hand-engineered tasks (OpenAI, 2023), ChatGPT-4 (chatgpt-4o-latest), intended for evaluation, and GPT-4 omni (gpt-4o-2024-08-06), OpenAI's most advanced flagship model[3].

Table 3 presents the results of the quality evaluation conducted by the meta-evaluators on the five hallucinated or faithful statements generated for each of the 2,170 test examples, alongside the rationalized evidence provided for each statement. The verification score was binary: a score of 1 was given if both the statement and the rationalized evidence were valid, and 0 if either was problematic. The evaluation results show that even the lowest-scoring domain, History, achieved a high-quality level, with 98.94% of its hallucinated statements deemed valid. Among the low-quality statements, 68% were flagged by two or more models, and 21% were flagged by all three models.

We employed three human annotators, all native Korean speakers, and graduates of four-year universities located in Seoul, Republic of Korea. The annotators reviewed 158 test examples flagged by at least one model as containing low-quality statements. They performed binary classification to determine whether revisions were necessary. A statement was considered for direct revision if two or more annotators agreed on the need for revision. As a result of the human evaluation, 137 test examples flagged as low-quality by the LLM meta-evaluators were found to reflect instances where the models either misinterpreted instructions or hallucinated during the evaluation process. For 21 test examples, at least two human annotators agreed that revisions were necessary. These problematic statements were subsequently revised by one of the authors, a native Korean speaker and Ph.D. candidate. Krippendorff's alpha for inter-annotator reliability was 0.923 among the three meta-evaluators and 0.828 among the three human annotators, indicating high and moderate inter-annotator agreement, respectively (Krippendorff, 2011).

## 4 EXPERIMENTS

### 4.1 SETUP

**Models** To ensure a broad representation of the LLMs employed in our experiments, we select models that have demonstrated strong performance within the open-source Korean NLP community and have been widely used as baselines in subsequent research or industries. We conduct experiments across three different model types: (1) open-source multilingual LLMs, including Llama2 (meta-llama/Llama-2-7b-chat-hf) (Touvron et al., 2023), Llama3 (meta-llama/Meta-Llama-3-8B-Instruct) (AI@Meta, 2024), and Mistral-Nemo (mistralai/Mistral-Nemo-Instruct-2407) (Jiang et al., 2023; MistralAI, 2024), (2) Korean-centric LLMs, such as KULLM3 (nlpai-lab/KULLM3) (Kim et al., 2024), which has been solely fine-tuned with limited instruction tuning dataset, and ExaOne (LGAI-EXAONE/EXAONE-3.0-7.8B-Instruct) (Research et al., 2024), which have undergone pre-training and instruction-tuning on broader dataset. These five open-source LLMs are capable of generating high-quality outputs based on the provided instructions with long-form documents and calculating log probabilities for our K-HALU evaluation framework. However, closed APIs, including GPT-3.5 Turbo (gpt-3.5-turbo-0125) (Ouyang et al., 2022), GPT-4 Turbo (gpt-4-turbo-2024-04-09), and GPT-4 omni (gpt-4o-2024-08-16) (OpenAI, 2023) do not provide access to log probabilities. As a result, evaluation for closed API models is limited to directly generating the indices of the statement choices.

---

[3]https://platform.openai.com/docs/models

**Log Probabilities**   We adopt multiple choice and log probabilities-based performance measurement as the default approach to ensure stability in performance reproduction and minimize unintended interference in evaluating hallucination detection capabilities. To evaluate the multiple-choice in K-HALU, we compute the conditional probabilities of sequence generation, leveraging the auto-regressive of LLMs (Gao et al., 2024). For each statement (choice) $x$ and its input source $S$, the sequence generation probability $P(x)$ is calculated as follows:

$$P(x) = \frac{1}{|x|} \sum_{i=1}^{|x|} \log \mathbb{P}(x_i | S : x_{<i})$$

(1)

Here, $P(x)$ represents the token probability computed by the model, with ":" indicating sequence concatenation. The input source content $S$ consists of the instruction $I$, the textual document $t$, and the publish date $d$, which can be expressed as $S = [I{:}t{:}d]$. For each of the 5 choices, the cumulative log probabilities of tokens are calculated independently by concatenating the input source $S$ with each choice $x$. Finally, the **Top-$N$ answers**, corresponding to the number of correct answers for the task, are selected based on their probabilities. This approach minimizes unintended interference from other choices and mitigates score distortion caused by differences in choice length and structure, maintaining stability in performance reproduction.

**Exact Match**   To enable LLMs to provide judgments directly, we set up the task to have the models generate binary outputs ("0" or "1") indicating the validity of each choice and evaluate them using exact match. We incorporate a post-processing step to minimize errors caused by unnecessary special characters or slight variations in format during the generation process. e.g., "[1, 1, 0, 0, 0]".

**LLM-as-a-Judge**   We use an LLM-as-a-Judge style prompt (See Table 13) to assign scores for hallucination detection results, enabling the LLM to evaluate the quality and accuracy of generated text (Liu et al., 2023; Chiang & Lee, 2023; Zheng et al., 2024). LLMs are tasked with directly generating faithful or hallucinated statements based on the given instructions. These outputs tend to be descriptive, which increases the potential for errors when evaluated using an exact match. To mitigate this issue, we utilize GPT-4 omni (`gpt-4o-2024-08-06`) as an evaluator to classify the validity of the generated statements as binary values (0 for invalid, 1 for valid).

## 4.2   RESULTS

**Baseline Accuracy**   Figure 2 presents the performance of baseline models on the K-HALU test set. Open-source LLMs exhibit an overall low performance, with an average accuracy of 29.05%. ExaOne achieves the highest accuracy at 32.95%, while Llama2 shows the lowest performance at 24.75%. Given that the model sizes of the open-source LLMs used in the experiment range between 7B and 13B, the relatively newer models, ExaOne and Mistral-Nemo, demonstrate better performance. Although KULLM3 is a Korean-specific LLM, it demonstrates limited improvement in detecting hallucinations in Korean, likely attributable to the tuning processes constrained by a limited dataset. In comparison, open-source LLMs demonstrate a substantial performance gap of 27.91% when compared to the average performance of 56.96% achieved by closed API LLMs[4], excluding GPT-4 omni, which is directly involved in dataset creation. These results highlight the vulnerability of open-source LLMs to hallucinations compared to their commercial counterparts. Proprietary LLMs exhibit inherent weaknesses in hallucination detection, stemming from constrained resources and incomplete tuning strategies. Even state-of-the-art closed API models, whether directly or indirectly involved in the dataset creation process, demonstrate suboptimal performance in detecting hallucinations in Korean, indicating a clear need for further enhancement.

**Domain Analysis**   Table 4 compares the baseline performance across the seven domains. Among the open-source LLMs, the highest average accuracy of 31.8% is observed in the Culture domain, while the lowest performance is seen in the Society domain, with an average accuracy of 27.06%. In contrast, closed API models show the highest average performance in the Technology domain

---

[4]Closed API LLMs are analyzed and compared with open-source LLMs using the exact match measurement as the default setup, given the restrictions on accessing log probabilities.

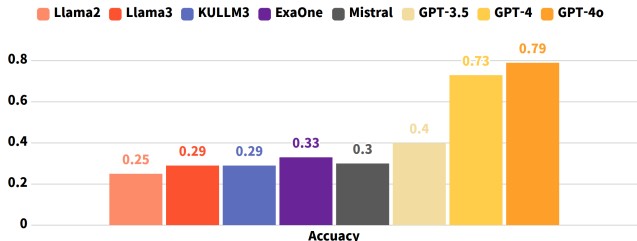

Figure 2: Baseline accuracy of models on the K-HALU test set. Open-source LLMs show lower accuracy, with ExaOne performing the best among them, while GPT-4 omni achieves the highest overall performance in comparison to other models.

Table 4: Model performance across seven knowledge domains, with accuracy evaluated using log probabilities. **Bold** indicates the domain where each model achieved the highest performance, while underline represents the domain where each model recorded the lowest performance.

| Models | Culture | Economy | History | International | Medical | Society | Technology |
|---|---|---|---|---|---|---|---|
| Llama2 (Touvron et al., 2023) | 0.2800 | 0.2642 | 0.2533 | 0.2219 | 0.2185 | 0.2177 | **0.2833** |
| Llama3 (AI@Meta, 2024) | **0.3200** | 0.2843 | 0.2733 | 0.3100 | 0.2923 | 0.2618 | 0.2733 |
| KULLM3 (Kim et al., 2024) | **0.3333** | 0.2876 | 0.2833 | 0.2736 | 0.2862 | 0.2744 | 0.2667 |
| ExaOne (Research et al., 2024) | 0.3367 | 0.3077 | 0.3033 | **0.3526** | 0.3385 | 0.3186 | 0.3467 |
| Mistral-Nemo (MistralAI, 2024) | **0.3200** | 0.3110 | 0.2933 | 0.3100 | 0.3077 | 0.2808 | 0.2867 |
| GPT-3.5 Turbo (Ouyang et al., 2022) | 0.3800 | 0.4013 | 0.3533 | **0.4316** | 0.4154 | 0.4164 | 0.4300 |
| GPT-4 Turbo (OpenAI, 2023) | **0.7667** | 0.7157 | 0.6967 | 0.7325 | 0.7384 | 0.7476 | 0.7433 |
| GPT-4 omni (OpenAI, 2023) | 0.7867 | 0.7659 | 0.7633 | 0.7994 | 0.7938 | 0.7950 | **0.8000** |

at 65.77%, and the lowest in the History domain at 60.44%. ExaOne exhibits similar performance variations to GPT-3.5, while the remaining open-source LLMs show comparable trends in their strengths and weaknesses across the domains. However, the overall performance variation across domains is not pronounced for any models. This suggests that the general-purpose models used in the experiment are relatively unaffected by domain-specific knowledge gaps or biases toward particular knowledge domains.

**Multiple Answers** Table 5 presents the baseline performance according to the number of correct answers across each domain. Open-source LLMs achieve the highest average accuracy of 31.67% for single-answer types while showing the lowest performance of 24.63% for questions requiring two correct answers. Models that perform well on single-answer types also tend to exhibit higher performance on more than two-answer types. Notably, there is a sharp decline in performance when inferring a single-answer in the Technology domain, two in the Society domain, and three in the History domain. These results suggest that the textual sources in the Society and History domains, which involve complex events, may have confused hallucination detection. Additionally, ExaOne's approximately 10% outperforming over other models in the Technology domain for single-answer questions suggests that the lack of training on technology-related knowledge written in Korean likely influenced the results. On the other hand, closed API models demonstrate a marked increase in performance on more than two-answer types. This outcome appears to indicate that these models recognized the correlation between answer candidates during the generative evaluation process or that the marginal differences among candidates in the hallucination or faithful sets resulted in higher scores when selecting multiple answers, regardless of the ranking of generation probabilities. This result supports the effectiveness of post-hoc sampling methods, such as the one proposed by Manakul et al. (2023), in mitigating hallucinations for closed API models.

**Instruction Types** Table 6 compares the performance based on whether the instruction type involves selecting hallucinated or faithful statements. All models, except GPT-3.5 Turbo, perform better when selecting faithful statements. Open-source LLMs exhibit a significant performance gap between the two question types, averaging 33.18%. In contrast, the closed API models show a marginal performance difference, averaging only 2.06%. These results indicate that open-source multilingual and Korean-centric LLMs still struggle to detect hallucinated outputs, raising concerns that their instruction tuning seems biased towards selecting faithful statements. Moreover, as the number of correct answers increases, performance on question types that require selecting hallucinated state-

Table 5: Model performance across seven domains based on the number of multiple answers with accuracy evaluated using log probabilities. (1/2/3) indicates the model's performance according to the number of labels. **Bold**: highest, underline: lowest performance.

| Models | Culture (1/2/3) | Economy (1/2/3) | History (1/2/3) | International (1/2/3) | Medical (1/2/3) | Society (1/2/3) | Technology (1/2/3) | Total (1/2/3) |
|---|---|---|---|---|---|---|---|---|
| Llama2 | 0.32 / 0.20 / 0.30 | 0.30 / 0.21 / 0.23 | 0.29 / 0.21 / 0.13 | 0.28 / 0.15 / 0.09 | 0.26 / 0.15 / 0.19 | 0.26 / 0.13 / 0.22 | 0.28 / 0.30 / 0.23 | **0.28** / 0.19 / 0.20 |
| Llama3 | 0.36 / 0.26 / 0.30 | 0.32 / 0.22 / 0.23 | 0.32 / 0.23 / 0.23 | 0.33 / 0.28 / 0.31 | 0.31 / 0.26 / 0.32 | 0.31 / 0.18 / 0.25 | 0.29 / 0.23 / 0.27 | **0.32** / 0.24 / 0.26 |
| KULLM3 | 0.38 / 0.26 / 0.30 | 0.31 / 0.27 / 0.20 | 0.33 / 0.24 / 0.10 | 0.29 / 0.24 / 0.25 | 0.30 / 0.25 / 0.32 | 0.35 / 0.13 / 0.25 | 0.26 / 0.26 / 0.33 | **0.32** / 0.23 / 0.25 |
| ExaOne | 0.35 / 0.30 / 0.37 | 0.30 / 0.32 / 0.30 | 0.35 / 0.27 / 0.13 | 0.33 / 0.37 / 0.44 | 0.31 / 0.36 / 0.45 | 0.37 / 0.22 / 0.31 | 0.38 / 0.31 / 0.27 | **0.34** / 0.31 / 0.33 |
| Mistral | 0.35 / 0.27 / 0.30 | 0.34 / 0.24 / 0.17 | 0.34 / 0.24 / 0.17 | 0.32 / 0.27 / 0.38 | 0.31 / 0.30 / 0.35 | 0.33 / 0.19 / 0.25 | 0.27 / 0.30 / 0.33 | **0.32** / 0.26 / 0.30 |
| GPT-3.5 | 0.20 / 0.69 / 0.53 | 0.30 / 0.56 / 0.53 | 0.26 / 0.58 / 0.23 | 0.30 / 0.63 / 0.59 | 0.25 / 0.69 / 0.58 | 0.27 / 0.62 / 0.69 | 0.27 / 0.64 / 0.73 | 0.27 / 0.63 / **0.56** |
| GPT-4 | 0.62 / 0.97 / 1.0 | 0.59 / 0.87 / 1.0 | 0.57 / 0.87 / 0.90 | 0.62 / 0.88 / 0.97 | 0.61 / 0.92 / 0.97 | 0.65 / 0.88 / 0.91 | 0.62 / 0.94 / 0.90 | 0.61 / 0.90 / **0.95** |
| GPT-4o | 0.66 / 0.98 / 1.0 | 0.66 / 0.89 / 1.0 | 0.64 / 0.93 / 0.97 | 0.69 / 0.96 / 1.0 | 0.67 / 0.98 / 0.97 | 0.70 / 0.93 / 0.97 | 0.69 / 0.97 / 0.97 | 0.67 / 0.95 / **0.98** |

Table 6: Model performance in hallucination and fact selection types across the number of multiple answers with accuracy evaluated using log probabilities. **H_Selection** refers to the hallucination selection type and **F_Selection** refers to the fact selection type. A number in () represents the number of labels, and _**Avg.** denotes the average score for each type. **Bold**: highest performance.

| Models | H_Selection (1) | F_Selection (1) | H_Selection (2) | F_Selection (2) | H_Selection (3) | F_Selection (3) | H_Avg. | F_Avg. |
|---|---|---|---|---|---|---|---|---|
| Llama2 | 0.1656 | 0.4015 | 0.0736 | 0.3089 | 0.0373 | 0.3611 | 0.1253 | **0.3696** |
| Llama3 | 0.1825 | 0.4554 | 0.0859 | 0.3884 | 0.0280 | 0.4815 | 0.1382 | **0.4378** |
| KULLM3 | 0.1656 | 0.4723 | 0.0644 | 0.4006 | 0.0186 | 0.4815 | 0.1207 | **0.4516** |
| ExaOne | 0.1488 | 0.5338 | 0.0644 | 0.5505 | 0.0467 | 0.6019 | 0.1134 | **0.5456** |
| Mistral | 0.1626 | 0.4815 | 0.0767 | 0.4465 | 0.0467 | 0.5463 | 0.1253 | **0.4774** |
| GPT-3.5 | 0.3098 | 0.2215 | 0.5920 | 0.6697 | 0.4860 | 0.6296 | 0.4120 | 0.3972 |
| GPT-4 | 0.6043 | 0.6231 | 0.8834 | 0.9266 | 0.9626 | 0.9352 | 0.7235 | **0.7456** |
| GPT-4o | 0.6288 | 0.7185 | 0.9479 | 0.9480 | 0.9813 | 0.9815 | 0.7594 | **0.8138** |

ments drops sharply. However, performance on faithful statement selection tends to improve. As discussed in *Multiple Answers*, this is considered to result from the smaller differences among candidates within the faithful set, whereas the larger dissimilarities within the hallucinated set arise from the model's inability to accurately identify hallucinations.

**Few-shot and CoT**  To address open-source LLMs' low hallucination detection capabilities, we apply few-shot samples and Chain-of-Thought (CoT) reasoning (Wei et al., 2022) to evaluate performance improvements. We create three samples containing multiple-answer types to measure few-shot accuracy. These samples are constructed based on the results from Table 6, with two selecting hallucinated statements and one selecting faithful statements. Additionally, we incorporate CoT reasoning steps, guiding the models to choose appropriate statements and rationalize their choices based on the provided documents and publication dates according to the instruction type. Appendix F illustrates the design of the CoT prompts. Table 7 compares the performance of the baseline models with the results after applying few-shot and CoT. Llama2 and ExaOne exhibit a slight decrease in performance, while Llama3, KULLM3, and Mistral show modest gains within 2%. These findings align with the results of HaluEval (Li et al., 2023), suggesting that few-shot sampling and CoT reasoning steps are not sufficient as fundamental solutions for improving hallucination detection.

**Exact Match and LLM-as-a-Judge**  Table 8 presents the results of applying evaluation methods where models generate answers directly based on the prompt's instruction and context. When using exact match to evaluate performance, it becomes more challenging for models to achieve high accuracy compared to multiple choice accuracy. This is because exact match heavily relies on instruction-following abilities, which amplifies performance differences among models. Additionally, the influence of instruction-following appears to have a greater impact on performance in 3-shot or CoT settings compared to other evaluation methods. To mitigate potential distortions caused by descriptive outputs in exact match evaluations, we employ the LLM-as-a-Judge evaluation method. Baselines show low performance as they struggle to generate answer choices that align with the given instructions. Interestingly, KULLM3 performs better in the LLM-as-a-Judge evaluation compared to other methods. ExaOne and Mistral-NEMO demonstrate generally higher performance across the multiple choice, exact match, and LLM-as-a-Judge evaluations. Similar to multiple choice and log-probability-based evaluations, applying 3-shot or CoT reasoning results in partial performance improvements. However, consistent performance enhancement is not observed across all settings. These results highlight the challenges open-source LLMs face in directly solving K-HALU's hallucination detection tasks through generative outputs or CoT prompting. This implies the need for further research to improve model capabilities in hallucination detection tasks effectively.

Table 7: Model performance by # shots and CoT with accuracy evaluated using log probabilities.

| Models | Zero-shot | 3-shot | 3-shot + CoT |
|---|---|---|---|
| Llama2 (Touvron et al., 2023) | **0.2475** | 0.2410 | 0.2429 |
| Llama3 (AI@Meta, 2024) | 0.2880 | 0.2880 | **0.2926** |
| KULLM3 (Kim et al., 2024) | 0.2862 | **0.3018** | 0.2959 |
| ExaOne (Research et al., 2024) | **0.3295** | 0.2922 | 0.2991 |
| Mistral-Nemo (MistralAI, 2024) | 0.3014 | 0.3060 | **0.3115** |

Table 8: Performance comparison of models using Exact Match and LLM-as-a-Judge.

| Models | Exact Match | | | LLM-as-a-Judge | | |
|---|---|---|---|---|---|---|
| | Zero-shot | 3-shot | 3-shot + CoT | Zero-shot | 3-shot | 3-shot + CoT |
| Llama2 (Touvron et al., 2023) | 0.0106 | 0.0051 | 0.0060 | 0.0101 | 0.0115 | 0.0189 |
| Llama3 (AI@Meta, 2024) | 0.1203 | 0.2396 | 0.1760 | 0.0484 | 0.0203 | 0.0212 |
| KULLM3 (Kim et al., 2024) | 0.0194 | 0.0083 | 0.0070 | **0.1134** | 0.1005 | 0.0797 |
| ExaOne (Research et al., 2024) | 0.0484 | 0.1885 | 0.1839 | 0.0922 | **0.1276** | 0.1023 |
| Mistral-NEMO (MistralAI, 2024) | **0.1240** | **0.2668** | **0.2770** | 0.0995 | 0.1138 | **0.1106** |

**Qualitative Analysis**  We qualitatively analyze cases where six or more models from the eight baselines select incorrect answers for a given example in each domain. Incorrect examples are categorized based on the rational evidence for faithfulness or hallucination in the answer options. We focus on the incorrect answer choices commonly identified across multiple models to summarize representative examples of hallucinations within the K-HALU benchmark. Figure 3 contains notes from our review of these incorrect examples across domains. Errors such as misrepresenting the publish date, misunderstanding numerical relationships, and conflicting knowledge are prevalent and recurrent issues across the models. In another example, models produce incorrect outputs for a faithful statement about 'NAVER's intelligent search robot announced in September 2000,' as they incorrectly believe the phrase "intelligent search robot" inappropriate. Likewise, when the document mentions 'the largest project in history,' models generate incorrect answers, arguing that this does not align with their prior knowledge. Also, models frequently produce domain-specific errors when trained on inaccurate knowledge of specific entities. For example, documents mentioning "President Syngman Rhee" consistently lead to incorrect answers across multiple models.

## 5    CONCLUSION

We propose the **K-HALU** benchmark, designed to evaluate hallucinations in Korean. K-HALU evaluates hallucination detection by verifying faithfulness using Korean knowledge-based textual documents drawn from seven different domains. The benchmark introduces a strict evaluation framework that includes multiple-answer questions, requiring models to select all possible correct answers. This approach enables us to quantify the hallucination detection capabilities of LLMs, providing deeper insight into their ability to discern faithfulness. Our analysis demonstrates that open-source LLMs are still struggling with hallucinations and that there remains a significant performance gap compared to closed API models. In future work, we aim to focus on enhancing datasets and models to mitigate hallucination in Korean, building on the insights gained from this research. We hope K-HALU helps improve the Korean NLP community's LLM hallucination performance.

## ETHICS STATEMENTS

The authors are solely responsible for any legal issues related to the K-HALU benchmark and acknowledge all associated risks. We state that the source dataset used to construct the K-HALU benchmark, *Knowledge Graph-to-Text*, was developed as part of the Intelligent Information Industry Infrastructure Construction project, supported by the Ministry of Science and ICT and the Korea National Information Society Agency (NIA). In utilizing this source dataset to create K-HALU, we comply with the Open AI Dataset (AI-Hub, S. Korea) copyright policy and confirm that its use and distribution have been pre-approved through prior consultation with NIA. The K-HALU benchmark is available for download and use by individuals who agree to the terms regarding its intended usage and the AI-Hub data usage policy. The link to access the dataset is provided in the K-HALU GitHub.

ACKNOWLEDGMENTS

K-HALU used datasets from The Open AI Dataset Project (AI-Hub, S. Korea). All dataset-related information can be accessed through AI-Hub (www.aihub.or.kr). This work was supported by Institute for Information & communications Technology Promotion (IITP) grant funded by the Korea government (MSIT) (RS-2024-00398115, Research on the reliability and coherence of outcomes produced by Generative AI). This research was supported by Basic Science Research Program through the National Research Foundation of Korea (NRF) funded by the Ministry of Education (NRF-2021R1A6A1A03045425). This work was supported by Institute of Information & communications Technology Planning & Evaluation (IITP) under the Leading Generative AI Human Resources Development (IITP-2024-R2408111) grant funded by the Korea government (MSIT)

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

## A    QUALITATIVE DETAILS

Figure 3 shows the analysis of hallucination error types frequently occurring in eight baselines across each domain.

| | ⚠ Error Categories | Examples |
|---|---|---|
| 🏛 Culture | 1. Knowledge Conflict 2. Publish Date 3. Inaccurate Knowledge of Entities 4. Lexical Ambiguity 5. Author Illusion | 1. Conflict with the latest knowledge regarding the adverb "most." 2. Miscalculation of the date in expressions like "this year." 3. Confusion regarding the impeachment order of the president. 4. Errors due to ambiguity in terms related to association membership. 5. Incorrect mapping of the book's author. |
| 📈 Economy | 1. Knowledge Conflict  2. Publish Date 6. Numerical Hallucination | 1. Judged the phrase "intelligent search robot" as inappropriate in the text. 2. Miscalculation of semesters related to "this year." 6. Error in calculating the bid amount. |
| ✏ History | 3. Inaccurate Knowledge of Entities 6. Numerical Hallucination 7. Negation | 3. Incorrect knowledge about the Korea historical incident. 6. Error in the event year. 7. Misinterpretation of negation. |
| 🌐 International | 1. Knowledge Conflict  2. Publish Date 4. Lexical Ambiguity 6. Numerical Hallucination | 1. Consistent errors regarding content related to Chinese censorship. 2. Errors concerning the publication date of a journal. 4. Over-generalization of the term "efficient." 6. Error in calculating the factory's revenue. |
| 💊 Medical | 1. Knowledge Conflict  2. Publish Date 3. Inaccurate Knowledge of Entities 6. Numerical Hallucination  7. Negation | 1. Knowledge conflict regarding terms like "latest" and "new drug." 2. Error in the event date. 3. Failure to distinguish between an academic conference and a forum. 6. Error in probability calculation. 7. Misinterpretation of negative expressions. |
| 👥 Society | 2. Publish Date 4. Lexical Ambiguity 5. Author Illusion | 2. Frequent errors regarding the publication year. 4. Ambiguity-related errors in interpreting educational content. 5. Spacing errors in the author's name. |
| 💻 Technology | 1. Knowledge Conflict  2. Publish Date 6. Numerical Hallucination 7. Negation | 1. Knowledge conflict about the new product. 2. Miscalculation of years related to "this year" and "last year." 6. Labeling error regarding the version of the game. 7. Misinterpretation of negative expressions. |

Figure 3: A review note for incorrect answers from baseline models.

## B    F1 SCORE

Table 9 presents the F1 score, precision, and recall metrics for evaluating model performance. These metrics are calculated using a macro-average approach, which considers the multiple correct answer distribution (6:3:1) and the uniform label distribution across the dataset. Accuracy requires selecting all correct answers to receive a score, meaning partially correct responses are treated as entirely incorrect. In contrast, the F1 score accommodates partial correctness, resulting in improved overall performance for the models.

Open-source LLMs, such as LLAMA, KULLM3, and ExaOne, tend to predict only a subset of the correct answers, prioritizing those with higher log probabilities. This behavior leads to higher precision compared to recall, as these models adopt a more conservative strategy in their predictions. On the other hand, closed API LLMs, including GPT-3.5, GPT-4, and GPT-4 omni, generate answers more flexibly, often producing a greater number of options than the predefined correct answers. This results in higher recall than precision, as these models aim to capture all possible correct answers but occasionally overpredict.

## C    EXTENDING K-HALU TO MULTILINGUAL

**Setup**    We conduct multilingual experiments using English (a high-resource language) and Malay (a low-resource language) to provide insights into the adaptability of the K-HALU benchmark across languages. K-HALU's hallucination detection criteria and settings for multiple-answer questions are designed to be language-independent, making the evaluation framework adaptable to different languages.

Table 9: Performance of LLMs on K-HALU evaluated by Accuracy, F1, Precision, and Recall.

| Model | Accuracy | F1 | Precision | Recall |
|---|---|---|---|---|
| Llama2 (Touvron et al., 2023) | 0.2475 | 0.3939 | 0.4152 | 0.3979 |
| Llama3 (AI@Meta, 2024) | 0.2880 | 0.4332 | 0.4503 | 0.4356 |
| KULLM3 (Kim et al., 2024) | 0.2862 | 0.4156 | 0.4323 | 0.4174 |
| ExaOne (Research et al., 2024) | **0.3295** | **0.4770** | **0.5117** | **0.4760** |
| Mistral-NEMO (MistralAI, 2024) | 0.3014 | 0.4393 | 0.4530 | 0.4417 |
| GPT-3.5 Turbo (Ouyang et al., 2022) | 0.4046 | 0.7237 | 0.6476 | 0.8260 |
| GPT-4 Turbo (OpenAI, 2023) | 0.7346 | 0.8538 | 0.8211 | 0.8899 |
| GPT-4 omni (OpenAI, 2023) | **0.7866** | **0.8732** | **0.8552** | **0.8924** |

Table 10: Performance of Models on K-HALU (English and Malay Versions).

| Models | Accuracy | F1 | Precision | Recall |
|---|---|---|---|---|
| **K-HALU (English ver.)** | | | | |
| Llama2 (Touvron et al., 2023) | 0.2757 | 0.4124 | 0.4261 | 0.4131 |
| Llama3 (AI@Meta, 2024) | 0.2727 | 0.4012 | 0.4139 | 0.4038 |
| KULLM3 (Kim et al., 2024) | 0.2846 | 0.4082 | 0.4235 | 0.4091 |
| ExaOne (Research et al., 2024) | 0.2638 | 0.4079 | 0.4288 | 0.4106 |
| Mistral-Nemo (MistralAI, 2024) | **0.2906** | **0.4354** | **0.4496** | **0.4364** |
| **K-HALU (Malay ver.)** | | | | |
| Llama2 (Touvron et al., 2023) | 0.2620 | 0.4317 | 0.4393 | 0.4314 |
| Llama3 (AI@Meta, 2024) | 0.2715 | 0.4443 | 0.4528 | 0.4452 |
| KULLM3 (Kim et al., 2024) | 0.2600 | 0.4268 | 0.4373 | 0.4274 |
| ExaOne (Research et al., 2024) | 0.2505 | 0.4171 | 0.4297 | 0.4189 |
| Mistral-Nemo (MistralAI, 2024) | **0.2772** | **0.4491** | **0.4591** | **0.4476** |

To set up K-HALU for multilingual evaluation, we translate 2,170 samples into English and Malay using the ChatGPT-4 (`chatgpt-4o-latest`) API. During the translation process, prompts are carefully crafted to minimize changes in hallucination-related content or linguistic nuances that could alter faithfulness (See Table 11). To ensure the quality of translated samples, we implement an LLM-as-a-Judge evaluation script using the GPT-4 omni (`gpt-4o-2024-08-06`) API to verify whether the translations preserved the original labels. Low-quality samples that failed this binary classification check are filtered out (See Table 12). As a result, 671 English samples and 523 Malay samples remain, indicating that only approximately 27.51% of the samples retain their validity after translation.

**Results** Table 10 presents the results of hallucination detection evaluation performed on the K-HALU benchmark using translations of the Korean textual documents and statements into English and Malay. The results indicate that LLMs continue to struggle with hallucination detection, even when K-HALU is evaluated in different languages. The Korean version of K-HALU achieves approximately 2.7% higher performance compared to the translated versions, highlighting the influence of language-specific embedded knowledge on hallucination detection performance.

To achieve optimal evaluation outcomes, it is essential to reconstruct reliable knowledge documents and faithful statements based on the original source in each language. Each language's cultural context and linguistic nuances require adjustments to hallucination detection methods and multiple-answer question approaches to ensure accuracy and relevance.

# D  LENS OBSERVATION

We employ two-lens observation methods to examine how each layer of LLM calculates the next token probability for both hallucinated and faithful statements.

Table 11: Prompt for translating K-HALU to English and Malay. This table demonstrates the instruction and system setup for translation tasks.

| |
|---|
| **### Translation Task Prompt — Korean to English ###** |
| **##System**: |
| You are a human annotator and a Korean-English translator. |
| Your task is to translate the provided Korean text into English without altering its meaning or structure. |
| **##Instruction**: |
| The text consists of "id", "document", "date", "instruction", statements "1", "2", "3", "4", "5", and "label". |
| Your task is to: |
| - Translate the "document", "instruction", and statements ("1", "2", "3", "4", "5") from Korean to English. |
| **##Requirements**: |
| - Ensure no changes are made beyond accurate translation. |
| - Maintain the structure of the original text in your translation. |
| - **Generate the result in the form of a JSON dictionary identical to the Output Format.** |
| **##Output Format**: |
| {"id":"{id}","document":"{translated_document}","date":"{date}", |
| "instruction":"{translated_instruction}", |
| "1":"{translated_statement_1}","2":"{translated_statement_2}", |
| "3":"{translated_statement_3}","4":"{translated_statement_4}", |
| "5":"{translated_statement_5}","label": "{label}"} |
| **##Input Format**: |
| **### Translation Task Prompt — Korean to Malay ###** |
| **##System**: |
| You are a human annotator and a Korean-Malay translator. |
| Your task is to translate the provided Korean text into Malay without altering its meaning or structure. |
| **##Instruction**: |
| The text consists of "id", "document", "date", "instruction", statements "1", "2", "3", "4", "5", and "label". |
| Your task is to: |
| - Translate the "document", "instruction", and statements ("1", "2", "3", "4", "5") from Korean to Malay. |
| **##Requirements**: |
| - Ensure no changes are made beyond accurate translation. |
| - Maintain the structure of the original text in your translation. |
| - **Generate the result in the form of a JSON dictionary identical to the Output Format.** |
| **##Output Format**: |
| {"id":"{id}","document":"{translated_document}","date":"{date}", |
| "instruction":"{translated_instruction}", |
| "1":"{translated_statement_1}","2":"{translated_statement_2}", |
| "3":"{translated_statement_3}","4":"{translated_statement_4}", |
| "5":"{translated_statement_5}","label": "{label}"} |
| **##Input Format**: |

**Logit Lens** (nostalgebraist, 2020) maps the model's representation space to the vocabulary space for each token, leveraging the residual stream for this process. This method enables us to observe how the model's final output evolves according to token and layer positions. It is widely used for interpreting the internal representations of Transformer-based language models (Dar et al., 2023; Hanna et al., 2024).

**Tuned Lens** (Belrose et al., 2023) addresses some limitations of the Logit Lens, such as its difficulty in dealing with inconsistent readiness for final decoding across different positions. The Tuned Lens improves upon the Logit Lens by aligning intermediate layer outputs more closely with final predictions, facilitating the capture of more abstract and semantic information (Nanda et al., 2023; Jiang et al., 2024).

The selection of positions for measuring token-wise changes in hidden states across layers through lens observation depends on the experimental objectives (Azaria & Mitchell, 2023; Chen et al., 2024). In our analysis, we focus on the final token of the output to assess the model's judgment and the potential occurrence of hallucinations in response to the input.

**Results** To further analyze the results from *Instruction Types*, we track the model's internal state during next-token prediction using lens observation. Figure 4 illustrates the probability shift across each layer for Llama2, according to hallucinated and faithful statements. Under the Logit Lens (nostalgebraist, 2020), slight probability fluctuations are detected in the early layers, with a gradual increase starting from the middle layer ($15^{th}$), and a sharp rise of approximately 40% near the final layer. For instructions that differentiate hallucinated statements, there is a slightly higher probability than for instructions that differentiate faithful statements up until the final layer, where the model becomes more confident about faithful statements. Under the Tuned Lens (Belrose et al., 2023), significant probability fluctuations are detected in the early layers, followed by another notable increase around the $15^{th}$ layer. After this point, little to no probability shifts are observed. Similar to the Logit

Table 12: Prompt for evaluating translation quality and hallucination label correctness. This table outlines the system and task instructions for assessment.

### Translation Evaluation Prompt — English Label Verification ###

**##System**:
You are a professional annotator tasked with evaluating the quality of translations in a dataset and ensuring the correctness of hallucination-related labels. Your role is to strictly assess whether the provided "label" aligns with the corresponding "document" and "instruction."

**##Task**:
Evaluate the following JSON object for its correctness based on the criteria below:
1. **Label Validity**: Verify if the provided "label" correctly reflects the instruction of each statement compared to the "document" and "instruction."

**##Requirements**:
- Do not provide explanations or justifications for your scores.
- Only output the score, e.g., 1 or 0.

**##Scoring**:
- Assign **1 point** if the "label" is valid.
- Assign **0 points** if the "label" is invalid.

**##Example Input**:
```
{
  "id": "001",
  "document": "The National Assembly is the legislative body of South Korea.",
  "date": "2024-11-01",
  "instruction": "Identify which statements are factually consistent with the document.",
  "1": "The National Assembly is located in Seoul.",
  "2": "It has 300 members elected by the public.",
  "3": "The Assembly also oversees the judiciary.",
  "4": "It was established in 1948.",
  "5": "The President of South Korea is a member of the National Assembly.",
  "label": [1,2,4]
}
```

**##Example Output**:
1

**##Input to Evaluate**:

### Translation Evaluation Prompt — Malay Label Verification ###

**##System**:
You are a professional annotator tasked with evaluating the quality of Malay translations in a dataset and ensuring the correctness of hallucination-related labels. Your role is to strictly assess whether the provided "label" aligns with the corresponding "document" and "instruction."

**##Task**:
Evaluate the following JSON object for its correctness based on the criteria below:
1. **Label Validity**: Verify if the provided "label" correctly reflects the instruction of each statement compared to the "document" and "instruction."

**##Requirements**:
- Do not provide explanations or justifications for your scores.
- Only output the score, e.g., 1 or 0.

**##Scoring**:
- Assign **1 point** if the "label" is valid.
- Assign **0 points** if the "label" is invalid.

**##Example Input**:
```
{
  "id": "001",
  "document": "Majlis Perundangan Kebangsaan adalah badan perundangan Korea Selatan.",
  "date": "2024-11-01",
  "instruction": "Kenal pasti pernyataan yang konsisten dengan fakta dalam dokumen.",
  "1": "Majlis Perundangan Kebangsaan terletak di Seoul.",
  "2": "Ia mempunyai 300 ahli yang dipilih oleh rakyat.",
  "3": "Majlis ini juga menyelia badan kehakiman.",
  "4": "Ia ditubuhkan pada tahun 1948.",
  "5": "Presiden Korea Selatan adalah ahli Majlis Perundangan Kebangsaan.",
  "label": [1,2,4]
}
```

**##Example Output**:
1

**##Input to Evaluate**:

Lens results, the model shows greater confidence in differentiating faithful statements compared to hallucinated ones as it approaches the final layer.

These findings suggest that open-source LLMs exhibit stronger certainty for faithful statements in their final layer outputs, contributing to the performance gap when handling instructions that differ-

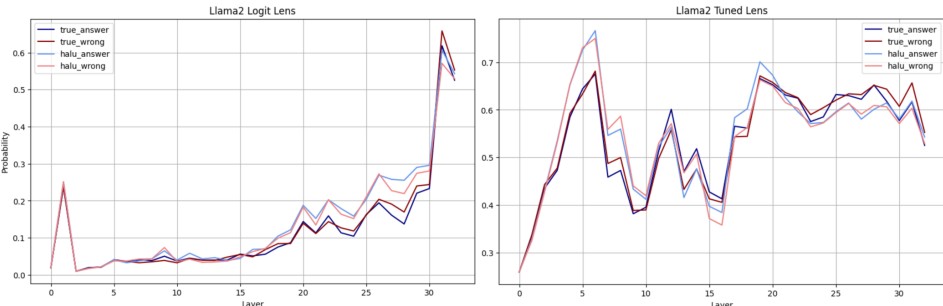

Figure 4: Token probability shift of Llama2 on the K-HALU under Logit and Tuned Lens. **true/halu** indicates that the instruction used selects faithful/hallucinated statements. **answer/wrong** refers to cases where the correct/incorrect option is chosen according to the instruction.

entiate hallucinated statements. The probability shift near the final layer reaffirms the importance of this stage in hallucination detection, as highlighted by Jiang et al. (2024) and Chen et al. (2024), and serves as a key factor explaining the performance difference observed in Table 6. Moreover, the Logit and Tuned Lens observations show that the internal token probabilities do not exhibit significant differences when instructions involve correct and incorrect statements. This indicates that the Llama2 model struggles to accurately assess hallucinated statements, which explains its notably poor performance in our previous experiments.

## E    LLM-AS-A-JUDGE STYLE PROMPT

Table 13 presents the prompt used for the LLM-as-a-Judge evaluation.

## F    CHAIN OF THOUGHT (COT) PROMPT

Table 14 presents the CoT prompt used for evaluating log probabilities, Table 15 for evaluating Exact matches, and Table 16 for the LLM-as-a-Judge evaluation method.

## G    K-HALU EXAMPLE

Figure 5, 6, 7, 8, 9, 10, and 11 illustrate examples of the original Korean texts from the K-HALU dataset for each domain.

Table 13: Prompt for evaluating LLMs' prediction quality and hallucination label correctness. This table outlines the system and task instructions for assessment.

### **Prediction Evaluation Prompt — Label Verification ###**

**##System**:

You are a professional annotator tasked with evaluating the quality of predictions in a dataset and ensuring the correctness of hallucination-related labels.

**##Task**:

Evaluate the following JSON object for its correctness based on the criteria below:

1. **Prediction Validity**: Verify if the provided "preds" correctly reflects the instruction of each statement compared to the "document" and "instruction."

**##Requirements**:

- Do not provide explanations or justifications for your scores.
- Only output the score, e.g., 1 or 0.

**##Scoring**:

- Assign **1 point** if the "preds" is valid.
- Assign **0 points** if the "preds" is invalid.

**##Example Input**:

```
{
  "id": "001",
  "document": "The National Assembly is the legislative body of South Korea.",
  "date": 2024-11-01
  "instruction": "Identify which statements are factually consistent with the document",
  "choices": [
    "1. The National Assembly is located in Seoul.",
    "2. It has 300 members elected by the public.",
    "3. The Assembly also oversees the judiciary.",
    "4. It was established in 1948.",
    "5. The President of South Korea is a member of the National Assembly."
  ],
  "gold": [
    "1. The National Assembly is located in Seoul.",
    "2. It has 300 members elected by the public.",
    "4. It was established in 1948."
  ],
  "preds": [
    "1. The National Assembly is located in Seoul.",
    "2. It has 300 members elected by the public."
  ]
}
```

**##Example Output**:

0

**##Input to Evaluate**:

Table 14: CoT reasoning prompt used for hallucination detection tasks evaluated using log probabilities. Each example illustrates the evaluation process, where statements are analyzed for consistency with the given document and classified as hallucinated or faithful.

### CoT Reasoning Prompt — Hallucination Detection (Log probabilities) ###

**##Example 1**:
date: {INPUT DATE}
document: {INPUT DOCUMENT}
instruction: {INPUT INSTRUCTION}
choice:
1. {#1 INPUT STATEMENT}
analysis: {REASON WHY} conclusion: {HALLUCINATED OR FAITHFUL}
2. {#2 INPUT STATEMENT}
analysis: {REASON WHY} conclusion: {HALLUCINATED OR FAITHFUL}
3. {#3 INPUT STATEMENT}
analysis: {REASON WHY} conclusion: {HALLUCINATED OR FAITHFUL}
4. {#4 INPUT STATEMENT}
analysis: {REASON WHY} conclusion: {HALLUCINATED OR FAITHFUL}
5. {#5 INPUT STATEMENT}
analysis: {REASON WHY} conclusion: {HALLUCINATED OR FAITHFUL}
output: {LIST OF CORRECT STATEMENTS}

**##Example 2**:
date: {INPUT DATE}
document: {INPUT DOCUMENT}
instruction: {INPUT INSTRUCTION}
choice:
1. {#1 INPUT STATEMENT}
analysis: {REASON WHY} conclusion: {HALLUCINATED OR FAITHFUL}
2. {#2 INPUT STATEMENT}
analysis: {REASON WHY} conclusion: {HALLUCINATED OR FAITHFUL}
3. {#3 INPUT STATEMENT}
analysis: {REASON WHY} conclusion: {HALLUCINATED OR FAITHFUL}
4. {#4 INPUT STATEMENT}
analysis: {REASON WHY} conclusion: {HALLUCINATED OR FAITHFUL}
5. {#5 INPUT STATEMENT}
analysis: {REASON WHY} conclusion: {HALLUCINATED OR FAITHFUL}
output: {LIST OF CORRECT STATEMENTS}

**##Example 3**:
date: {INPUT DATE}
document: {INPUT DOCUMENT}
instruction: {INPUT INSTRUCTION}
choice:
1. {#1 INPUT STATEMENT}
analysis: {REASON WHY} conclusion: {HALLUCINATED OR FAITHFUL}
2. {#2 INPUT STATEMENT}
analysis: {REASON WHY} conclusion: {HALLUCINATED OR FAITHFUL}
3. {#3 INPUT STATEMENT}
analysis: {REASON WHY} conclusion: {HALLUCINATED OR FAITHFUL}
4. {#4 INPUT STATEMENT}
analysis: {REASON WHY} conclusion: {HALLUCINATED OR FAITHFUL}
5. {#5 INPUT STATEMENT}
analysis: {REASON WHY} conclusion: {HALLUCINATED OR FAITHFUL}
output: {LIST OF CORRECT STATEMENTS}

Table 15: CoT reasoning prompt used for hallucination detection tasks evaluated using exact match. Each example illustrates the evaluation process, where statements are analyzed for consistency with the given document and classified as hallucinated or faithful.

### CoT Reasoning Prompt — Hallucination Detection (Exact Match) ###

##Example 1:
date: {INPUT DATE}
document: {INPUT DOCUMENT}
instruction: {INPUT INSTRUCTION}
choice:
1. {#1 INPUT STATEMENT}
analysis: {REASON WHY} conclusion: {HALLUCINATED OR FAITHFUL} $\rightarrow$ {0 $\vee$ 1}
2. {#2 INPUT STATEMENT}
analysis: {REASON WHY} conclusion: {HALLUCINATED OR FAITHFUL} $\rightarrow$ {0 $\vee$ 1}
3. {#3 INPUT STATEMENT}
analysis: {REASON WHY} conclusion: {HALLUCINATED OR FAITHFUL} $\rightarrow$ {0 $\vee$ 1}
4. {#4 INPUT STATEMENT}
analysis: {REASON WHY} conclusion: {HALLUCINATED OR FAITHFUL} $\rightarrow$ {0 $\vee$ 1}
5. {#5 INPUT STATEMENT}
analysis: {REASON WHY} conclusion: {HALLUCINATED OR FAITHFUL} $\rightarrow$ {0 $\vee$ 1}
output: {[BINARY VECTOR OF 5 ELEMENTS]}

##Example 2:
date: {INPUT DATE}
document: {INPUT DOCUMENT}
instruction: {INPUT INSTRUCTION}
choice:
1. {#1 INPUT STATEMENT}
analysis: {REASON WHY} conclusion: {HALLUCINATED OR FAITHFUL} $\rightarrow$ {0 $\vee$ 1}
2. {#2 INPUT STATEMENT}
analysis: {REASON WHY} conclusion: {HALLUCINATED OR FAITHFUL} $\rightarrow$ {0 $\vee$ 1}
3. {#3 INPUT STATEMENT}
analysis: {REASON WHY} conclusion: {HALLUCINATED OR FAITHFUL} $\rightarrow$ {0 $\vee$ 1}
4. {#4 INPUT STATEMENT}
analysis: {REASON WHY} conclusion: {HALLUCINATED OR FAITHFUL} $\rightarrow$ {0 $\vee$ 1}
5. {#5 INPUT STATEMENT}
analysis: {REASON WHY} conclusion: {HALLUCINATED OR FAITHFUL} $\rightarrow$ {0 $\vee$ 1}
output: {[BINARY VECTOR OF 5 ELEMENTS]}

##Example 3:
date: {INPUT DATE}
document: {INPUT DOCUMENT}
instruction: {INPUT INSTRUCTION}
choice:
1. {#1 INPUT STATEMENT}
analysis: {REASON WHY} conclusion: {HALLUCINATED OR FAITHFUL} $\rightarrow$ {0 $\vee$ 1}
2. {#2 INPUT STATEMENT}
analysis: {REASON WHY} conclusion: {HALLUCINATED OR FAITHFUL} $\rightarrow$ {0 $\vee$ 1}
3. {#3 INPUT STATEMENT}
analysis: {REASON WHY} conclusion: {HALLUCINATED OR FAITHFUL} $\rightarrow$ {0 $\vee$ 1}
4. {#4 INPUT STATEMENT}
analysis: {REASON WHY} conclusion: {HALLUCINATED OR FAITHFUL} $\rightarrow$ {0 $\vee$ 1}
5. {#5 INPUT STATEMENT}
analysis: {REASON WHY} conclusion: {HALLUCINATED OR FAITHFUL} $\rightarrow$ {0 $\vee$ 1}
output: {[BINARY VECTOR OF 5 ELEMENTS]}

Table 16: CoT reasoning prompt used for hallucination detection tasks evaluated using LLM-as-a-Judge. Each example illustrates the evaluation process, where statements are analyzed for consistency with the given document and classified as hallucinated or faithful.

### CoT Reasoning Prompt — Hallucination Detection (LLM-as-a-Judge) ###

**##Example 1**:
date: {INPUT DATE}
document: {INPUT DOCUMENT}
instruction: {INPUT INSTRUCTION}
choice:
1. {#1 INPUT STATEMENT}
analysis: {REASON WHY} conclusion: {HALLUCINATED OR FAITHFUL} $\rightarrow$ {0 $\vee$ 1}
2. {#2 INPUT STATEMENT}
analysis: {REASON WHY} conclusion: {HALLUCINATED OR FAITHFUL} $\rightarrow$ {0 $\vee$ 1}
3. {#3 INPUT STATEMENT}
analysis: {REASON WHY} conclusion: {HALLUCINATED OR FAITHFUL} $\rightarrow$ {0 $\vee$ 1}
4. {#4 INPUT STATEMENT}
analysis: {REASON WHY} conclusion: {HALLUCINATED OR FAITHFUL} $\rightarrow$ {0 $\vee$ 1}
5. {#5 INPUT STATEMENT}
analysis: {REASON WHY} conclusion: {HALLUCINATED OR FAITHFUL} $\rightarrow$ {0 $\vee$ 1}
output: {LIST OF CORRECT STATEMENTS}

**##Example 2**:
date: {INPUT DATE}
document: {INPUT DOCUMENT}
instruction: {INPUT INSTRUCTION}
choice:
1. {#1 INPUT STATEMENT}
analysis: {REASON WHY} conclusion: {HALLUCINATED OR FAITHFUL} $\rightarrow$ {0 $\vee$ 1}
2. {#2 INPUT STATEMENT}
analysis: {REASON WHY} conclusion: {HALLUCINATED OR FAITHFUL} $\rightarrow$ {0 $\vee$ 1}
3. {#3 INPUT STATEMENT}
analysis: {REASON WHY} conclusion: {HALLUCINATED OR FAITHFUL} $\rightarrow$ {0 $\vee$ 1}
4. {#4 INPUT STATEMENT}
analysis: {REASON WHY} conclusion: {HALLUCINATED OR FAITHFUL} $\rightarrow$ {0 $\vee$ 1}
5. {#5 INPUT STATEMENT}
analysis: {REASON WHY} conclusion: {HALLUCINATED OR FAITHFUL} $\rightarrow$ {0 $\vee$ 1}
output: {LIST OF CORRECT STATEMENTS}

**##Example 3**:
date: {INPUT DATE}
document: {INPUT DOCUMENT}
instruction: {INPUT INSTRUCTION}
choice:
1. {#1 INPUT STATEMENT}
analysis: {REASON WHY} conclusion: {HALLUCINATED OR FAITHFUL} $\rightarrow$ {0 $\vee$ 1}
2. {#2 INPUT STATEMENT}
analysis: {REASON WHY} conclusion: {HALLUCINATED OR FAITHFUL} $\rightarrow$ {0 $\vee$ 1}
3. {#3 INPUT STATEMENT}
analysis: {REASON WHY} conclusion: {HALLUCINATED OR FAITHFUL} $\rightarrow$ {0 $\vee$ 1}
4. {#4 INPUT STATEMENT}
analysis: {REASON WHY} conclusion: {HALLUCINATED OR FAITHFUL} $\rightarrow$ {0 $\vee$ 1}
5. {#5 INPUT STATEMENT}
analysis: {REASON WHY} conclusion: {HALLUCINATED OR FAITHFUL} $\rightarrow$ {0 $\vee$ 1}
output: {LIST OF CORRECT STATEMENTS}

**K-HALU: Culture Domain**

{"id": "p30ts_235",
"document": "이중섭(1916~1956)은 한국전쟁과 가난을 피해 일본인 아내와 두 아들을 도쿄 처가에 보낸 후 다시 만날 날을 애타게 기다렸다. 아내가 보낸 편지를 벽에 붙여두고 치열하게 그림만 그렸다. 그러나 1956년 조현병과 간염으로 병원을 전전하다 세상을 떠났고, 결국 뼛가루만 가족 품에 안겼다. \n\n1956년 이중섭이 죽음을 앞두고 그린 마지막 그림 '돌아오지 않는 강'은 일본에 있는 아내와 북한에 남아 생사를 확인하지 못한 어머니에 대한 그리움을 절절하게 쏟아부었다. 그림에서 창문 밖 휘몰아치는 강물을 바라보는 남자 얼굴에 수심이 가득하다. 현해탄을 건너 아내를 다시 만날 수 없다는 절망 때문이다. \n\n이 유화가 올해 케이옥션 첫 경매에 나온다. 가로 14.6㎝, 세로 18.5㎝로 작품 크기가 작아서 추정가는 1억5000만~3억원이다. 케이옥션은 22일 오후 4시 서울 강남구 신사동 본사에서 이중섭 그림을 비롯해 100억원 규모 미술품 172점을 출품한다고 밝혔다. \n\n이번에 여성 블루칩 작가 천경자(1924~2015)의 1982년 작 '꽃을 든 여인'도 추정가 7억~12억원에 새 주인을 찾는다. 화려한 꽃다발을 들었지만 매섭고 공허한 눈빛을 가진 여인을 그린 작품이다. 긴 목으로 고독을 뿜어내는 이 여인은 작가 분신이기도 하다. 아름다움에 대한 갈망과 오랜 한(恨)을 동시에 품은 것 같다. \n\n고미술 부문에서 가장 눈에 띄는 작품은 퇴계 이황(1501~1570) 등 조선시대 주요 인물들 간찰(편지)을 모은 '고간독(古柬牘)'이다. 16~19세기 주요 인물 159명의 간찰 180점과 행주 기씨 집안 간찰 13점 등 간찰 총 193점이 9책에 나뉘 수록돼 있다. 퇴계 이황과 고봉 기대승(1527~1572) 간의 '사단칠정논변'과 관련된 서간(書簡)이 가장 시선을 붙잡는다. '퇴계선생문집' 권16, '답기명언', '고봉전서' 중 '양선생왕복서' 권1에 수록된 이 서간들은 8년에 걸친 '사단칠정논변'의 시작을 알리는 글이다. 정탁, 한호, 이덕형, 최명길, 김육, 김상헌, 송시열, 남구만, 이재 등 조선시대 정치·경제·사상·문화사에 깊은 족적을 남긴 이들의 간찰도 수록돼 있다. 조선 중기와 후기에 걸친 서예사 흐름을 파악할 수 있는 이 간독집 추정가는 9000만~2억원이다. \n\n최근 미술전문매체 아트넷이 지난 10년간 세계 미술시장에서 가격이 많이 상승한 작가 100명에 선정한 이우환(21위), 박서보(53위), 정상화(90위) 작품도 나왔다. \n\n이우환 1985년 작 '동풍 S.8508B'는 추정가 16억~23억원에, 2008년 작 'Dialogue(대화)'는 5억~6억원에 출품된다. 1970년대 점·선 시리즈부터 1980년대 이후 바람 시리즈, 1990년대 조응 시리즈에 이르기까지 다양한 시리즈를 제작한 이우환 그림 가격은 한국 생존 작가 중에서 가장 비싸다. 미국 뉴욕 구겐하임미술관, 워싱턴 허시혼미술관 야외조각공원, 프랑스 퐁피두 메츠 센터 등에서 개인전을 열면서 세계 미술계 중심에 올라섰다. \n\n박서보의 2005년 작인 개나리색 '묘법 No. 050615'는 2500만~5000만원에 새 주인을 찾는다. 1970년대부터 지금까지 한국 단색화를 이끌고 있는 그는 시대에 따라 변화하는 '묘법'을 선보여 왔다. 신체 움직임을 통해 고유의 정신세계를 표현하는 묘법 시리즈는 한 폭의 명상 작품으로 평가받는다. \n\n푸른 빛을 발하는 정상화 1987년 작 '무제 87 7 A'는 추정가 3억8000만~6억원에 나왔다. 현재 뉴욕 레비고비 갤러리에서 개인전을 열고 있는 작가는 세계 미술계에서 집중 조명을 받고 있다. 미술시장 부동의 1위를 차지하고 있는 김환기(1913~1974) 작품 7점도 출품된다. 자연에의 합일 과정이 드러나는 'VI 68', 1968년 뉴욕의 추상적 하늘 풍경을 확인할 수 있는 '메아리 I 24 III 68 #4', 1960년대 후반부터 구상적인 형상들이 사라지고 스며드는 물감들이 만들어낸 색채가 두드러지는 작품 '1 IIII 69#49' 등이 출품된다. \n\n경매 출품작을 직접 확인할 수 있는 경매 프리뷰는 11일 시작해 22일까지 진행된다. 사전 예약 없이 누구나 무료로 관람이 가능하다.",
"date": "20200110",
"instruction": "주어진 문서를 통해 파악할 수 있는 내용에 해당하는 문장을 선택하시오. 단, 정답 문장은 여러 개 존재할 수 있다.",
"1": "이우환은 다양한 시리즈의 작품을 제작하며 세계 미술계에서 높은 평가를 받고 있다.",
"2": "이우환의 작품 '동풍 S.8508B'는 2021년 경매에서 50억 원에 낙찰되었다.",
"3": "이중섭의 마지막 작품 '돌아오지 않는 강'은 그의 아내와 함께 그린 공동 작품이다.",
"4": "퇴계 이황과 고봉 기대승의 '사단칠정논변'은 20세기 초에 작성된 서간이다.",
"5": "이중섭의 마지막 작품 '돌아오지 않는 강'은 그의 개인적인 비극과 절망을 반영하고 있다.",
"label": [0, 4],
"choice_number": [1, 5]}

Figure 5: An example of K-HALU in Culture domain

---

**K-HALU: Economy Domain**

{"id": "p60hs_82",
"document": "GS그룹은 2005년 1월 LG그룹과 분리돼 세워졌다. LG그룹의 모태는 1947년 설립된 락희화학공업인데, 구인회 회장과 허만정 씨가 공동 창업주였다. 두 가문은 1947년부터 2005년까지 분쟁 없이 운영해오다 분할한 것이다. \n\nGS그룹의 주력 사업은 GS칼텍스로 대표되는 에너지와 GS리테일로 잘 알려진 유통업이다. GS그룹의 총 계열사는 69개이지만 상장사는 6개에 불과해 기업 공개율은 8.7%에 그친다. 주력 기업인 GS칼텍스마저도 현재는 비상장사이기 때문이다보니 재계 평균인 16.6%보다 낮다. GS그룹 내부거래비율은 5.5%로 30대 그룹 평균인 8.1%보다 낮은 수준이지만 2013년 3.3%를 기록한 후 상승 추세에 있는 것이 특징이다. \n\n수익 구조를 보면 지주회사인 GS의 당기순이익은 상장 계열사 중엔 GS리테일이, 비상장계열사 중엔 GS칼텍스 기여도가 절대적이다. 2015년 기준으로 상장사 6곳 중 GS글로벌을 제외한 나머지 5개사의 단순평균 ROE(자기자본이익률)는 5.9%다. GS 당기순이익에 기여도가 가장 큰 상장 계열사가 GS리테일(GS지분 65.7%)로 비중이 21.4%나 된다. 그러나 실질적으로 가장 큰 기여를 하는 계열사는 GS가 지분 50%를 보유한 비상장계열사 GS칼텍스다. \n\n즉 관건은 그룹 최대 계열사인 GS칼텍스의 기업공개 여부다. 이 회사가 상장될 경우 그룹 지배구조와 그룹 내 계열사 주주가치에도 긍정적인 영향을 미칠 가능성이 크기 때문이다. \n\nGS칼텍스의 지분구조는 미국 석유회사인 셰브런(Chevron)이 지분 50%를 보유하고 있으며, GS가 100% 소유한 GS에너지가 나머지 50%를 보유 중이다. 따라서 셰브런의 동의 없이는 기업공개는 불가능한 상황이다. 참고로 GS칼텍스의 자산총액은 GS 자산총액과 유사한 수준이다. \n\n지주회사인 GS의 현금배당성향은 높은 편이다. 이는 업종 특성상 자회사의 현금배당이 주 매출액으로, 2012년부터 2015년까지 자회사로부터 유입된 현금배당금이 393억원, 2033억원, 1381억원, 644억원이었다. \n\n안상희 대신지배구조연구소 연구위원은 \"자회사로부터 거둬들인 현금배당금이 변동이 컸지만 이 기간 GS는 평균 1279억원의 현금배당금을 유지했다\"며 \"지주회사란 특성 외에도 주주구성상 지배주주 등 친족 지분율이 42.8%로 높았기 때문\"이라고 분석했다. \n\n지주회사 체제를 구축하고 있는 GS그룹 내부지분율은 69.55%로 재계 평균(60.62%)보다 높은 편이다. 그러다보니 지배구조는 안정적인 편이다. \n\n주목할 점은 최대주주 특수관계인이 49명에 달한다는 점이다. 홍자 돌림의 4세 경영체제 중심을 앞두고 있는 상황에서 지배구조에 대한 논의가 필요할 것으로 보인다. \n\n상장계열사 중 GS, GS건설, 삼양통상의 최대주주는 계열사가 아닌 다수의 친족들로 구성돼 있다. 특히 GS건설, 삼양통산, 승산 등 일부 계열사는 GS그룹의 지배주주(허창수)와는 다른 친족관계인 점을 고려하면 향후 지배구조 변화 시 관심이 필요하다. \n\n또 하나 주목할 점은 총수 일가의 등기임원 등재율이 34.8%로 재계 평균인 21.1% 대비 높은 수준이란 것이다. 이는 총수일가 숫자가 타 그룹 대비 많다는 것이 주요 요인으로 뽑힌다. GS의 경우 총수 일가의 수가 49명에 달한다. \n\n등기임원이 과도하게 겸임하고 있어 사내이사로서 충실한 임무수행이 어려울 수 있다는 점은 해결해야 할 과제다. GS의 정택근 대표이사가 8개를 겸직하고 있으며 삼양통상 허광수 대표이사(8개), GS홈쇼핑의 정찬수 이사(6개)도 과도하게 겸직하고 있다. \n\n특히 겸직 중엔 계열사 감사까지 맡고 있는 경우가 있어 독립성 확보 측면에서 논란의 여지가 있다. 삼양통상의 허남각 대표가 경원건설 감사를, GS홈쇼핑의 유경수 사내이사가 GS텔레서비스 등 4개 계열사를 감사하고 있다. 정찬수 이사 역시 GS칼텍스 등 4개 계열사 감사를 맡고 있다.",
"date": "20170419",
"instruction": "주어진 문서의 내용과 다르거나 불확실한 환각 문장을 고르시오. 단, 환각 문장은 여러 개 존재할 수 있다.",
"1": "GS그룹의 총수 일가의 등기임원 등재율은 재계 평균보다 높다.",
"2": "GS칼텍스의 상장은 셰브런의 동의가 필요하다.",
"3": "GS그룹의 내부거래비율은 상승 추세에 있다.",
"4": "GS그룹의 상장 계열사 중 GS리테일이 당기순이익 기여도가 가장 크다.",
"5": "GS그룹의 주력 사업은 GS리테일로 대표되는 에너지와 GS칼텍스로 잘 알려진 유통업이다.",
"label": [4],
"choice_number": [5]}

Figure 6: An example of K-HALU in Economy domain

---

**K-HALU: History Domain**

{"id": "p30ts_233",
"document": "명성황후는 청나라와 러시아를 활용한 능수능란한 외교술로 한반도를 병합하려는 일본의 야욕을 번번이 좌절시켰다. 그러나 외세에 의존한 외교는 한계가 분명했다. 결국 1895년 10월 8일 새벽 경복궁에서 을미사변으로 불리는 전대미문의 사건이 일어난다. 조선 주재 일본 공사 미우라 고로가 지휘하는 일본 낭인들에게 명성황후가 시해된 것이다.",
"date": "20200320",
"instruction": "주어진 문서를 통해 파악할 수 있는 내용에 해당하는 문장을 선택하시오. 단, 정답 문장은 여러 개 존재할 수 있다.",
"1": "을미사변은 1895년 10월 8일 저녁 경복궁에서 일어났다.",
"2": "을미사변은 1895년 10월 8일 새벽 창덕궁에서 일어났다.",
"3": "을미사변은 일본 공사 미우라 고로가 지휘한 일본 낭인들에 의해 명성황후가 시해된 사건이다.",
"4": "명성황후는 청나라와 러시아를 외교적으로 활용하여 일본의 한반도 병합 시도를 저지했다.",
"5": "조선 주재 일본 공사 미우라 고로가 지휘하는 청나라 낭인들에게 명성황후가 시해된 것이다.",
"label": [3, 2],
"choice_number": [4, 3]}

---

Figure 7: An example of K-HALU in History domain

---

**K-HALU: International Domain**

{"id": "p60hs_4",
"document": "일본이 이르면 내년 1월 인도와 원자력 협정을 체결하고 원전 수출을 위한 기반을 닦는다. 또 아프리카에 에너지·광물자원 개발을 위해 20억달러를 투자하는 등 국외 자원과 인프라스트럭처 시장 개척에 공격적으로 나선다.\n\n니혼게이자이신문은 20일 일본이 이르면 내년 1월 인도와 원자력 협정을 체결하고 원전 수출을 모색한다고 보도했다.\n\n아베 신조 총리는 이달 27~30일 일본을 방문하는 만모한 싱 인도 총리와 정상회담을 하면서 2011년 후쿠시마 제1원전 사고로 중단했던 원자력 협정 체결 교섭을 재개한다는 데 합의할 방침이다. \n\n양국은 이르면 내년 1월 원자력 협정을 체결할 예정으로 아베 총리가 직접 인도를 방문해 협정에 서명하는 방안도 논의되고 있다. 일본이 인도와 원자력 협정에 적극 나서는 것은 인도가 2020년까지 100조원 가까운 돈을 들여 원전 18기를 증설하기 때문이다.\n\n특히 인도는 원전사고 발생 시 사업자뿐 아니라 원자로 제작사에도 소송을 제기하기 때문에 미국 업체들이 수주를 꺼리고 있어 일본 도시바 히타치 미쓰비시중공업 등이 수주할 가능성이 높다고 신문은 전했다.\n\n신문은 \"독자 기술을 고집하는 중국이나 러시아 원전시장과 달리 인도시장은 일본 기술력을 과시할 수 있는 기회라고 생각해 정부가 적극 나서고 있다\"고 설명했다.\n\n일본은 또 지난주 아프리카 15개국 자원담당 대표자들과 J-서밋 콘퍼런스를 하고 에너지·광물 자원 개발에 향후 5년간 20억 달러(약 2조2300억원)를 투자할 계획이라고 18일 발표했다. \n\n모테기 도시미쓰 경제산업상은 \"이번 콘퍼런스는 일본 기업들에 대해 아프리카 투자를 독려하고 아프리카가 안정적인 성장을 이루는 데 도움을 주기 위한 것\"이라고 말했다. \n\n일본은 아프리카에서 원유, 천연가스, 철광석 등 천연 광물을 장기간 안정적으로 공급받고 도로, 철도, 전력 등 각종 인프라스트럭처 건설에 참여하는 방안을 추진하고 있다.\n\n아베 총리는 또 6월 요코하마에서 개최되는 제5차 아프리카개발회의(TICAD) 참석차 일본을 방문하는 아프리카 40개국 정상들과 릴레이 정상회담도 할 예정이다.",
"date": "20130521",
"instruction": "주어진 문서의 내용과 다르거나 불확실한 환각 문장을 고르시오. 단, 환각 문장은 여러 개 존재할 수 있다.",
"1": "아베 총리는 인도와의 원자력 협정 체결을 위해 직접 인도를 방문할 가능성이 있다.",
"2": "일본은 아프리카에서 원유, 천연가스, 철광석 등의 천연 광물을 안정적으로 공급받기 위해 노력하고 있다.",
"3": "일본은 아프리카에 에너지와 광물 자원 개발을 위해 20억 달러를 투자할 예정이다.",
"4": "일본은 내년 1월 인도와 원자력 협정을 체결하고, 동시에 인도에 50억 달러를 투자할 계획이다.",
"5": "일본은 후쿠시마 원전 사고 이후 중단되었던 원자력 협정 교섭을 재개할 계획이다.",
"label": [3],
"choice_number": [4]}

---

Figure 8: An example of K-HALU in International domain

| K-HALU: Medical Domain |
|---|
| {"id": "p60hs_70",
"document": "담배를 피거나 술을 마시고 뚱뚱한 노인일수록 낙상·욕창 등 노인증후군을 앓을 확률이 더 높은 것으로 나타났다. 6일 국민건강보험공단은 대한노인병학회와 공동 연구를 통해 그같은 결과를 도출했다고 밝혔다. \n\n연구진은 2006~2015년 건강보험 빅데이터를 기반으로 4대 노인증후군인 낙상 관련 골절과 섬망(과잉행동·초조감), 실금(대·소변을 무의식적으로 배출), 욕창으로 진단받은 65세 이상 노인 135만여 명을 대상으로 위험인자를 분석했다. \n\n그 결과 비만은 실금을 1000명당 16.1명가량 발생시키며 위험도도 1.3배 높이는 것으로 나타났다. 흡연을 하면 낙상 관련 골절은 1000명당 6.4명으로 1.47배 더 많이 나타나고 욕창은 1000명당 13.2명으로 위험도를 1.25배 높이는 것으로 조사됐다. 주 3회 이상 음주를 하면 낙상 관련 골절은 1.05배, 섬망은 1.13배 높게 나타났다. 5가지 이상 약물을 복용하는 경우에도 낙상 관련 골절이 1.64배, 욕창은 1.69배 더 많이 발생하는 것으로 밝혀졌다. \n\n노인증후군 발생을 예방하는 데에는 운동이 효과적이라는 결론도 나왔다. 운동을 자주 하면 낙상 관련 골절은 20% 줄어들고 섬망과 실금, 욕창도 각각 17%와 7%, 25%씩 감소하는 것으로 나타났다. \n\n노인증후군을 앓는 환자의 동반질환을 살핀 결과 역시 치매와 긴밀한 상관관계를 지니는 것으로 조사됐다. 동반질환 중 치매 환자는 낙상 관련 골절이 2.74배, 섬망은 1.32배, 실금 1.5배, 욕창 2.9배가량 더 많이 나타났다. 뇌졸중과 신장질환, 골다공증 같은 만성질환도 노인증후군과 상관성이 높은 것으로 확인됐다. 특히 여성은 남성과 비교해 섬망과 실금이 각각 2.4배씩 더 많이 나타나는 것으로 드러났다. \n\n이번 연구를 총괄한 원장원 경희의료원 가정의학과 교수는 \"노인증후군은 요양시설 입소율과 사망위험을 함께 증가시킨다\"며 \"건강습관 개선을 통해 노인증후군 발생을 충분히 줄일 수 있다\"고 말했다.",
"date": "20181206",
"instruction": "주어진 문서의 내용과 다르거나 불확실한 환각 문장을 고르시오. 단, 환각 문장은 여러 개 존재할 수 있다.",
"1": "비만, 흡연, 음주가 노인증후군 발생 위험을 높인다.",
"2": "흡연은 낙상 관련 골절을 줄이고 욕창을 예방한다.",
"3": "치매는 노인증후군과 긴밀한 상관관계를 가진다.",
"4": "건강습관 개선은 노인증후군 발생을 줄일 수 있다.",
"5": "여성 노인은 남성 노인보다 섬망과 실금 발생률이 높다.",
"label": [1],
"choice_number": [2]} |

Figure 9: An example of K-HALU in Medical domain

| K-HALU: Society Domain |
|---|
| {"id": "p30hs_230",
"document": "교육당국이 인공지능(AI) 기술의 기초 원리를 가르치는 거점형 일반고를 육성한다. \n\n교육부는 올해 처음 도입되는 'AI 융합 교육과정 운영 고등학교'를 전국에 34곳 선정했다고 9일 밝혔다. AI 융합 교육과정 운영고는 내년 신입생부터 2023년까지 전체 교과 수업의 15%가량을 정보과학, 프로그래밍, 빅데이터 분석, 데이터 과학, 인공지능 수학(가칭) 등 AI 관련 과목으로 편성한다. 또한 이들 학교는 공동 교육과정을 1주일에 2시간 이상 개설하는 등 인근의 다른 학교 학생들에게 AI교육 이수 기회를 제공하는 거점 역할도 맡는다. \n\n이번에 AI 융합 교육과정 운영고로 선정된 학교는 서울 동양고·서라벌고·오산고·태릉고·환일고, 경기 김포제일고·매탄고·송내고·세교고·일산대진고, 인천 연송고·청라고 등이다. 비수도권 지역에서는 부산 동아고·삼정고, 대구 화원고·대건고, 광주 서강고, 대전 대고·대전여고, 울산 경의고, 강원 치악고, 충북 주성고, 충남 논산대건고·천안오성고·천안월봉고, 전남 무안고·문태고·순천매산고, 경북 안동고·안동중앙고·포항제철고, 경남 마산구암고·마산삼진고, 제주 중앙여고 등도 선정됐다. \n\n교육부는 향후 각 학교별로 4년간 예산 2억5000만원을 지원할 계획이다. 올해는 거점형 일반고에 학교당 1억원의 예산을 지원하고, 2021년부터 2023년까지는 매년 5000만원을 추가 지원할 예정이다. 교육부는 \"올해는 준비기로서, 학교는 정보교육실 등 창의·융합 교육을 위한 환경을 구축하고, 내년 신입생을 위한 교육과정을 준비한다"고 전했다. 이 과정에서 교육부는 인공지능 융합 과목에 대한 교사의 지도 역량을 강화하기 위해 여름·겨울 방학을 이용한 심화 연수 과정을 추진하며, 교육대학원(석사 학위 과정)을 통한 AI 관련 교육 전문인력도 양성할 계획이다. \n\n한편 올해 모든 초·중학교에는 소프트웨어(SW) 과목이 필수화된다. 교육부는 연내로 초·중·고 단계별 인공지능 교육 기준안을 마련할 예정이다.",
"date": "20200309",
"instruction": "주어진 문서의 내용과 다르거나 불확실한 환각 문장을 고르시오. 단, 환각 문장은 여러 개 존재할 수 있다.",
"1": "교육부는 AI 관련 교육 전문인력을 양성하기 위해 교육대학원 석사 학위 과정을 활용할 계획이다.",
"2": "교육부는 AI 융합 교육과정 운영 고등학교를 통해 인공지능 관련 과목을 전체 교과 수업의 15%로 편성할 계획이다.",
"3": "올해는 거점형 일반고에 학교당 10억원의 예산을 지원하고, 2021년부터 2023년까지는 매년 1억원을 추가 지원할 예정이다.",
"4": "교육부는 올해 처음 도입되는 'AI 융합 교육과정 운영 고등학교'를 전국에 340곳 선정했다고 9일 밝혔다.",
"5": "교육부는 AI 융합 교육과정 운영 고등학교에 4년간 총 2억5000만원의 예산을 지원할 계획이다.",
"label": [3, 2],
"choice_number": [4, 3]} |

Figure 10: An example of K-HALU in Society domain

---

**K-HALU: Technology Domain**

{"id": "p60hs_46",
"document": "인공지능(AI) 다국어 호텔 챗봇 '레드타이버틀러'를 개발한 레드타이가 다음달 서울 신림역 근처에 AI 호텔 '호텔 레드타이' 1호점을 선보인다. 레드타이는 이를 시작으로 AI 호텔 프랜차이즈 사업을 본격 시작한다. \n\n호텔 레드타이는 스마트폰, 키오스크, 챗봇, 사물인터넷 (IoT), 음성봇, 로봇 등을 통해 비대면 서비스를 선호하는 MZ세대를 주 고객으로 겨냥했다. MZ 세대는 1980년대 초~2000년대 초 출생한 밀레니얼 세대와 1990년대 중반~2000년대 초반 출생한 Z세대를 통칭하는 말로, 디지털 환경에 익숙하고 주체적인 소비를 지향하는 특징이 있다. \n\n특히 11월에 문을 열 예정인 호텔 레드타이 신림 1호점은 지상 13층, 지하 2층 53객실 규모 신축 호텔이다. 1층 로비는 셀프 체크인·체크아웃을 위한 안면인식 키오스크, 프런트에는 AI 아바타와 AI 음성봇이 24시간 고객을 응대한다. AI 무인 자판기를 이용한 무인 편의점과 다양한 소모임을 위한 공유 스페이스도 운영할 계획이다. 또 AI의 차가운 이미지를 상쇄하기 위해 식물을 주요 디자인 요소로 배치한 것이 특징이다. \n\n레드타이는 전략적 제휴를 맺고 있는 국내 각 분야 기업과 협업해 비대면 AI 호텔을 구축했다. 레드타이의 호텔 챗봇은 고객이 체크인하기 전부터 전반적으로 컨시어지를 담당한다. 야놀자는 클라우드 기반 객실 관리 솔루션인 와이플럭스로 셀프 체크인·체크아웃, 객실 정비 등 서비스 요청, 키리스(Keyless) 방식의 객실 출입, 실내 조명 ·온도 조절 등 객실 제어를 담당한다. \n\n레드타이는 중소형 호텔 사업을 하고 있거나, 준비 중인 예비 사업자들이 레드타이의 비대면 AI 호텔 솔루션을 도입함으로써, 운영 효율화와 수익 극대화에 도움을 받을 수 있을 것으로 기대했다. \n\n정승환 레드타이 대표는 \"코로나19 이후 안전과 청결, 비대면의 관광 고객 요구가 변해가고 비대면과 거리 두기가 일상화되면서, AI 호텔이 침체된 관광 숙박 시장에 긍정적 영향력을 전달할 것\"이라고 말했다.",
"date": "20201013",
"instruction": "주어진 문서의 내용과 다르거나 불확실한 환각 문장을 고르시오. 단, 환각 문장은 여러 개 존재할 수 있다.",
"1": "레드타이는 AI 호텔 프랜차이즈 사업을 본격적으로 시작할 계획이다.",
"2": "호텔 레드타이는 대면 서비스를 선호하는 MZ세대를 주요 고객으로 삼고 있다.",
"3": "레드타이는 80년대부터 00년대 초반의 세대들을 주요 타겟으로 한다.",
"4": "레드타이는 다양한 국내 기업과 협업하여 비대면 AI 호텔을 구축했다.",
"5": "호텔 레드타이 신림 1호점은 13층 규모의 신축 호텔이다.",
"label": [1],
"choice_number": [2]}

Figure 11: An example of K-HALU in Technology domain

