# OpenReview forum: "K-HALU: Multiple Answer Korean Hallucination Benchmark for Large Language Models"
_ICLR.cc/2025/Conference — ICLR 2025 Poster_

### Official Review · Reviewer_zS85 · 2024-10-30

**Soundness:** 4
**Presentation:** 4
**Contribution:** 3
**Rating:** 8
**Confidence:** 4

**Summary:**

This paper introduces K-HALU, a multiple-answer Korean hallucination benchmark for evaluating LLMs on their ability to detect factual inaccuracies or hallucinations within Korean-specific contexts. Unlike previous Korean benchmarks, which often rely on translations of English datasets or offer limited dataset access, K-HALU provides a culturally and linguistically relevant resource. It consists of 2,170 test samples. By requiring models to select all possible correct answers in a multiple-choice format, K-HALU presents a stricter evaluation standard than traditional benchmarks, especially for open-source models, which often struggle with hallucination detection.

**Strengths:**

The main strength of this paper lies in the originality and cultural relevance of K-HALU, as it fills a significant gap in the Korean NLP community by using carefully selected and annotated content rather than translations from existing benchmarks. K-HALU’s focus on Korean linguistic and socio-cultural contexts ensures that factuality assessments and hallucination detection are appropriately designed, making it particularly valuable for future NLP research in non-English settings. The multiple-answer format provides a more rigorous evaluation by requiring the selection of all correct answers, offering a strict measure of model robustness and enhancing the benchmark's reliability in assessing hallucination vulnerabilities in real-world applications. Additionally, figures in the paper reveal that open-source models still struggle with hallucinations in Korean, highlighting opportunities for further development and application in multilingual hallucination research.

**Weaknesses:**

A weakness of the paper is the lack of clarity regarding the human-annotated statements. The paper does not provide details about the annotators' backgrounds, such as their expertise in relevant domains (e.g., whether they hold advanced degrees or come from diverse educational backgrounds). This information is critical for understanding the quality and reliability of the annotations, especially in fields that require specialised knowledge. Secondly, the evaluation section appears unclear and lacks specificity. It is unclear whether exact match is used as the primary metric for assessing correctness in multiple-answer questions or if external models are used in evaluation.  A more detailed breakdown of the evaluation methods would improve clarity and provide stronger support for the paper’s conclusions.

**Questions:**

1. For common benchmarks like MMLU, the majority of samples can be sourced online to verify the correctness of extractions. Can K-HALU samples similarly be sourced online for users to verify correctness?

2. In evaluating multiple-choice answers (as mentioned in the weaknesses section), is exact match used as the criterion for correctness, or are alternative evaluation methods applied?

3. What would the model accuracy be if strict exact match criteria were applied, where an incorrect answer is not picked and the entire question being marked as incorrect? A breakdown of accuracy under these conditions for each model would provide valuable insights into model performance.

4. What would the feasibility of extending K-HALU to other non-English or low-resource languages be? Is the framework adaptable beyond Korean, or are there limitations specific to the Korean language context?

---

> ### Author Response · Authors · 2024-11-21
> **Response to Reviewer zS85 - Part 1**
>
> We sincerely thank Reviewer zS85 for recognizing the value of our work and for investing time in reviewing our manuscript. Below are our responses to the reviewer’s comments.
>
> **Q1. There is a lack of clarity regarding the human-annotated statements.**
>
> **R1.** We appreciate the reviewer's comment and have clarified the details of the human annotator consensus process (In 3.3 Quality Control; Ensemble Verification).
>
> **We have added the following content after lines 290-300**:
>
> *We employed three human annotators, all native Korean speakers, and graduates of four-year universities located in Seoul, Republic of Korea. The annotators reviewed 158 test examples flagged by at least one model as containing low-quality statements. They performed binary classification to determine whether revisions were necessary. A statement was considered for direct revision if two or more annotators agreed on the need for revision. As a result of the human evaluation, 137 test examples flagged as low-quality by the LLM meta-evaluators were found to reflect instances where the models either misinterpreted instructions or hallucinated during the evaluation process. For 21 test examples, at least two human annotators agreed that revisions were necessary. These problematic statements were subsequently revised by one of the authors, a native Korean speaker, and Ph.D. candidate. Krippendorff’s alpha for inter-annotator reliability was 0.923 among the three meta-evaluators and 0.828 among the three human annotators, indicating high and moderate inter-annotator agreement, respectively (Krippendorff, 2011).*
>
> **Q2. The evaluation section appears unclear and lacks specificity.**
>
> **R2.** For multiple-answer questions, we calculate the cumulative log probabilities for each choice token based on the given input document and instructions. The log probabilities are then exponentiated (using exp) to convert them into actual probabilities. The top N choices with the highest probabilities are selected as the correct answers. This approach does not require external models and is one of the commonly used methods in NLP benchmarks for evaluating LLMs using log probabilities [1, 2].
>
> [1] EleutherAI/lm-evaluation-harness
> https://github.com/EleutherAI/lm-evaluation-harness
>
> [2] Huggingface/Openllm leaderboard
> https://huggingface.co/spaces/open-llm-leaderboard-old/open_llm_leaderboard
>
> We have incorporated the reviewer's feedback (Q4) by adding Exact Match and LLM-as-a-Judge experiments to enable a more diverse comparison of evaluation methods. Along with these additions, **we have revised the 4.1 Setup section** to provide a clearer explanation of the log probability calculation method for the multiple-choice format, as outlined below:
>
> **Log probabilities**:
> We adopt multiple choice and log probabilities-based performance measurement as the default approach to ensure stability in performance reproduction and minimize unintended interference in evaluating hallucination detection capabilities. To evaluate the multiple-choice in K-HALU, we compute the conditional probabilities of sequence generation, leveraging the auto-regressive of LLMs (Gao et al., 2024). For a sequence $x$ and its input source $S$, the sequence generation probability $P(x)$ is calculated as:
>
> $P(x) = \frac{1}{|x|} \sum_{i=1}^{|x|} \log \mathbb{P}(x_{i} \mid S:x_{<i})$
>
> Here, $P(x)$ represents the token probability computed by the model, with ":" indicating sequence concatenation. The input source content $S$ consists of the instruction $I$, the textual document $t$, and the publish date $d$, which can be expressed as $S = [I\text{:}t\text{:}d]$. For each statement from the set of 5 choices, we calculate the cumulative log probabilities for its tokens. These log probabilities are then exponentiated to convert them into actual probabilities. The \textbf{Top-$N$ answers} with the highest probabilities are selected as the model's predicted answers.
>
> **Exact Match**: To enable LLMs to provide judgments directly, we set up the task to have the models generate binary outputs (“0” or “1”) indicating the validity of each choice and evaluate them using exact match. We incorporate a post-processing step to minimize errors caused by unnecessary special characters or slight variations in format during the generation process. e.g., “[1, 1, 0, 0, 0]”.
>
> **LLM-as-a-Judge**: We use an LLM-as-a-Judge style prompt to assign scores for hallucination detection results, enabling the LLM to evaluate the quality and accuracy of generated text (Liu et al., 2023; Chiang & Lee, 2023; Zheng et al., 2024). LLMs are tasked with directly generating faithful or hallucinated statements based on the given instructions. These outputs tend to be descriptive, which increases the potential for errors when evaluated using an exact match. To mitigate this issue, we utilize GPT-4 omni (gpt-4o-2024-08-06) as an evaluator to classify the validity of the generated statements as binary values (0 for invalid, 1 for valid)

---

> ### Author Response · Authors · 2024-11-21
> **Response to Reviewer zS85 - Part 2**
>
> **Q3. Can K-HALU samples similarly be sourced online for users to verify correctness?**
>
> **R3.** As noted in Footnote 2, the source data for K-HALU originates from datasets shared by the National Information Society Agency (NIA), a public government institution in South Korea, for research purposes. While the dataset does not include exact links to the original news articles, it provides essential metadata such as the publish date and the source entity. Additionally, all datasets are accompanied by metadata, authoring tools, and expert validation, ensuring their reliability compared to datasets collected by private entities or individuals. Since the data is derived from publicly accessible materials, users can easily cross-check the accuracy of the information through independent searches.
>
> **Q4. In evaluating multiple-choice answers. is exact match used as the criterion for correctness, or are alternative evaluation methods applied?**
>
> **R4.** Multiple-choice answers are evaluated differently from the Exact Match method. As explained in **R2**, the process involves summing the log probabilities of token generation for each choice based on the given input. The model then selects the choices in descending order of their cumulative probabilities, identifying those with the highest likelihood for the given query.
>
> We adopt the multiple-choice answers format to mitigate issues that could arise in generative evaluations, such as length imbalance, variability in sentence expressions, and structural differences that may distort performance. It also reduces numerical instability caused by summing log probabilities and ensures consistent evaluation scores. By minimizing unintended factors in performance assessment, we believe this approach provides a more valid comparison of hallucination detection performance across models.
>
> Moreover, the multiple-choice format has been widely adopted in benchmark datasets, as demonstrated by its use in Huggingface’s OpenLLM Leaderboard (https://huggingface.co/spaces/open-llm-leaderboard-old/open_llm_leaderboard). This method of comparing model performance using log probabilities and multiple-choice questions has been validated through thousands of model evaluations within the NLP community [1,2,3,4].
>
> [1] Zellers, R., Holtzman, A., Bisk, Y., Farhadi, A., & Choi, Y. (2019, July). HellaSwag: Can a Machine Really Finish Your Sentence?. In *Proceedings of the 57th Annual Meeting of the Association for Computational Linguistics* (pp. 4791-4800).
>
> [2] Clark, P., Cowhey, I., Etzioni, O., Khot, T., Sabharwal, A., Schoenick, C., & Tafjord, O. (2018). Think you have solved question answering? try arc, the ai2 reasoning challenge. *arXiv preprint arXiv:1803.05457*.
>
> [3] Hendrycks, D., Burns, C., Basart, S., Zou, A., Mazeika, M., Song, D., & Steinhardt, J. Measuring Massive Multitask Language Understanding. In *International Conference on Learning Representations*.
>
> [4] Lin, S., Hilton, J., & Evans, O. (2022, May). TruthfulQA: Measuring How Models Mimic Human Falsehoods. In *Proceedings of the 60th Annual Meeting of the Association for Computational Linguistics (Volume 1: Long Papers)* (pp. 3214-3252).

---

> ### Author Response · Authors · 2024-11-21
> **Response to Reviewer zS85 - Part 3**
>
> **Q5. What is the model accuracy under strict exact match criteria, where any incorrect answer marks the entire question as wrong?**
>
> **R5.** In the multiple-choice-based accuracy measurement, even a single incorrect choice is treated as a wrong answer. To board analysis, we have incorporated an F1 score that reflects partial correctness. Additionally, in response to the reviewer's feedback, we have introduced stricter generative evaluation methods, including **Exact Match** and **LLM-as-a-Judge** approaches (the setup for these methods was detailed in **R2**).
>
> **K-HALU with F1** (**Appendix B**)
> | Model          | Accuracy | F1    | Precision | Recall  |
> |----------------|----------|-------|-----------|---------|
> | LLAMA2         | 0.2475   | 0.3939| 0.4152    | 0.3979  |
> | LLAMA3         | 0.2880   | 0.4332| 0.4503    | 0.4356  |
> | KULLM3         | 0.2862   | 0.4156| 0.4323    | 0.4174  |
> | ExaOne         | 0.3295   | 0.477 | 0.5117    | 0.476   |
> | Mistral NEMO   | 0.3014   | 0.4393| 0.453     | 0.4417  |
> | GPT-3.5        | 0.4046   | 0.7237| 0.6476    | 0.826   |
> | GPT-4          | 0.7346   | 0.8538| 0.8211    | 0.8899  |
> | GPT-4o         | 0.7866   | 0.8732| 0.8552    | 0.8924  |
>
> Accuracy requires all correct answers to be selected for scoring, meaning partially correct answers are treated as incorrect. F1, however, accounts for partial correctness, leading to improved overall performance for models.
>
> **Exact Match Performance of LLMs** (**4.2 Results**)
> | Model          | 0-shot | 3-shot  | 3-shot + CoT |
> |----------------|-----------|---------|--------------|
> | LLAMA2         | 0.0106    | 0.0051  | 0.006        |
> | LLAMA3         | 0.1203    | 0.2396  | 0.176        |
> | KULLM3         | 0.0194    | 0.0083  | 0.007        |
> | ExaOne         | 0.0484    | 0.1885  | 0.1839       |
> | Mistral NEMO   | 0.124     | 0.2668  | 0.277        |
>
> When using exact match to evaluate performance, it becomes more challenging for models to achieve high accuracy compared to multiple-choice accuracy. This is due to the involvement of instruction-following abilities, which highlights more pronounced performance differences among models. Similar to evaluations based on multiple-choice and log probabilities, applying shots and CoT reasoning shows partial performance improvement. However, consistent performance enhancement is not observed across all settings.
>
> **LLM-as-a-Judge Performance of LLMs**  (**4.2 Results**)
> | Model          | Zero-shot | 3-shot  | 3-shot + CoT |
> |----------------|-----------|---------|--------------|
> | LLAMA2         | 0.0101    | 0.0115  | 0.0189       |
> | LLAMA3         | 0.0484    | 0.0203  | 0.0212       |
> | KULLM3         | 0.1134    | 0.1005  | 0.0797       |
> | ExaOne         | 0.0922    | 0.1276  | 0.1023       |
> | Mistral NEMO   | 0.0995    | 0.1138  | 0.1106       |
>
> To address potential distortions caused by descriptive outputs in exact match evaluations, we used GPT-4o to validate the generated results. Interestingly, KULLM3 performed better in this evaluation compared to other methods. Additionally, ExaOne and Mistral NEMO demonstrated generally higher performance across log probability, exact match, and LLM-as-a-Judge evaluations.
>
> These results highlight the challenges open-source LLMs face in directly solving K-HALU’s hallucination detection tasks through generative outputs or CoT prompting. This implies the need for further research to improve model capabilities in hallucination detection tasks effectively.

---

> ### Author Response · Authors · 2024-11-21
> **Response to Reviewer zS85 - Part 4**
>
> **Q6. The feasibility of extending K-HALU to other non-English or low-resource languages. Are there limitations specific to the Korean language context?**
>
> **R6.** Extending K-HALU to other non-English or low-resource languages is theoretically feasible. Key components of K-HALU, such as hallucination detection criteria and settings for multiple correct answers, are designed to be language-independent, making the framework adaptable to different languages.
>
> However, to achieve optimal evaluation outcomes, it is essential to reconstruct reliable knowledge documents and faithful statements based on the original source in each language. Each language's cultural context and linguistic nuances require adjustments to hallucination detection methods and multiple-answer question approaches to ensure accuracy and relevance.
>
> To address the reviewer's curiosity, we have conducted additional experiments using English (a high-resource language) and Malay (a low-resource language). These experiments provide insights into the adaptability of the K-HALU framework across languages and are detailed in the manuscript. These results and settings have been included in **Appendix C Extending K-HALU to Multilingual.**
>
> **Performance of Models on K-HALU (English and Malay Versions)**
>
> **K-HALU (English ver.)**
> | Model          | Accuracy | F1     | Precision | Recall  |
> |----------------|----------|--------|-----------|---------|
> | LLAMA2         | 0.2757   | 0.4124 | 0.4261    | 0.4131  |
> | LLAMA3         | 0.2727   | 0.4012 | 0.4139    | 0.4038  |
> | KULLM3         | 0.2846   | 0.4082 | 0.4235    | 0.4091  |
> | ExaOne         | 0.2638   | 0.4079 | 0.4288    | 0.4106  |
> | Mistral NEMO   | 0.2906   | 0.4354 | 0.4496    | 0.4364  |
>
> **K-HALU (Malay ver.)**
> | Model          | Accuracy | F1     | Precision | Recall  |
> |----------------|----------|--------|-----------|---------|
> | LLAMA2         | 0.2620   | 0.4317 | 0.4393    | 0.4314  |
> | LLAMA3         | 0.2715   | 0.4443 | 0.4528    | 0.4452  |
> | KULLM3         | 0.2600   | 0.4268 | 0.4373    | 0.4274  |
> | ExaOne         | 0.2505   | 0.4171 | 0.4297    | 0.4189  |
> | Mistral NEMO   | 0.2772   | 0.4491 | 0.4591    | 0.4476  |
>
> Tables present the results of the hallucination detection evaluation performed on the K-HALU benchmark using translations of the Korean textual documents and statements into English and Malay. The results indicate that LLMs continue to struggle with hallucination detection, even when K-HALU is evaluated in different languages. The Korean version of K-HALU achieves approximately 2.7\% higher performance compared to the translated versions, highlighting the influence of language-specific embedded knowledge on hallucination detection performance.
>
> We sincerely thank Reviewer zS85 for the thoughtful review and valuable feedback. We hope our detailed responses have effectively addressed your concerns.

---

> > ### Author Response · Authors · 2024-11-25
> > **Looking forward to your response**
> >
> > Dear Reviewer zS85,
> >
> > Thank you for your valuable time in reviewing our work and for providing constructive feedback. **We submitted our revised manuscript along with detailed responses to your comments a few days ago.** We would greatly appreciate it if you could share any additional thoughts or concerns to help us further improve and address any remaining issues.
> >
> > In our revisions, we carefully considered your suggestions and made the following changes:
> >
> > - As recommended, we added a detailed explanation of the human-annotated statements in **3.3 Quality Control, Lines 290-300**.
> >
> > - To improve clarity regarding evaluation, we expanded **4.1 Setup** with additional details and included configurations for **log probabilities, Exact Match, and LLM-as-a-Judge methods**.
> >
> > - Regarding the K-HALU samples, we clarified that their source data is publicly accessible and includes sufficient metadata (e.g., publication dates and publishers) to **enable cross-verification**, provided users comply with the terms of use.
> >
> > - We justified the use of **multiple-choice and log probabilities** as evaluation methods by referencing **recent studies, including the Open LLM leaderboard and lm-eval-harness**, to substantiate the validity and robustness of our approach.
> >
> > - To address your concerns about evaluation, we added experiments for **Exact Match and LLM-as-a-Judge** methods in **4.1 Setup** and **4.2 Results**.
> >
> > - To complement the strict evaluation approach involving multiple answers, we introduced the **F1 scoreas an additional metric**.
> >
> > - To assess the feasibility of extending K-HALU, we conducted additional experiments on **English (a high-resource language)** and **Malay (a low-resource language)**, which are now included in **Appendix C**.
> >
> > We have made every effort to thoughtfully incorporate your feedback and improve the manuscript. We sincerely hope that the revisions address your concerns and look forward to your kind reassessment.
> >
> > Best regards,
> >
> > The Authors

---

> > > ### Comment · Reviewer_zS85 · 2024-11-25
> > >
> > > Thanks for the clarifications — they were really helpful. While the explanations address all the doubts clearly, it would be extremely beneficial to include them in the camera-ready version, as addressing these doubts would strengthen the paper as well. Before reconsidering the scores, it would be helpful to have a bullet-point summary of how the authors plan to improve the script based on the feedback.

---

> > > > ### Author Response · Authors · 2024-11-25
> > > > **Plan to improve the script - part 1**
> > > >
> > > > We sincerely appreciate the reviewer’s feedback and are grateful for evaluation that our response effectively addressed concerns and was deemed extremely beneficial to the manuscript.
> > > >
> > > > The following comments **how each bullet point has addressed to revise and improve the manuscript**.
> > > >
> > > > > - As recommended, we added a detailed explanation of the human-annotated statements in **3.3 Quality Control, Lines 290-300**.
> > > >
> > > > &rarr; **Improvement**: Lines 290–300 include details about the nationality and educational background of the three human annotators, a comprehensive evaluation process, inter-annotator agreement results, and the nationality and educational background one of the authors responsible for the final review.
> > > >
> > > > **[Revision (Page 6, Lines 290 to 300)]**
> > > >
> > > > *We employed three human annotators, all native Korean speakers, and graduates of four-year universities located in Seoul, Republic of Korea. The annotators reviewed 158 test examples flagged by at least one model as containing low-quality statements. They performed binary classification to determine whether revisions were necessary. A statement was considered for direct revision if two or more annotators agreed on the need for revision. As a result of the human evaluation, 137 test examples flagged as low-quality by the LLM meta-evaluators were found to reflect instances where the models either misinterpreted instructions or hallucinated during the evaluation process. For 21 test examples, at least two human annotators agreed that revisions were necessary. These problematic statements were subsequently revised by one of the authors, a native Korean speaker and Ph.D. candidate. Krippendorff’s alpha for inter-annotator reliability was 0.923 among the three meta-evaluators and 0.828 among the three human annotators, indicating high and moderate interannotator agreement, respectively (Krippendorff, 2011).*
> > > >
> > > > > - To improve clarity regarding evaluation, we expanded **4.1 Setup** with additional details and included configurations for **log probabilities, Exact Match, and LLM-as-a-Judge methods**.
> > > >
> > > > &rarr; **Improvement**: The subsection titled *Evaluation* has been renamed to **Log Probabilities**, emphasizing that the default evaluation method is based on log probabilities for multiple-choice performance measurement. The description of the calculation method explicitly clarifies that the cumulative log probabilities of tokens for each choice are computed, and the Top-N answers are extracted based on these values. Additionally, explanations for the generative evaluation setups, Exact Match and LLM-as-a-Judge, have been included to provide a comprehensive understanding of the evaluation framework.
> > > >
> > > > **[Revision (Page 7, Lines 327 to 355)]**
> > > >
> > > > *Log probabilities:
> > > > We adopt multiple choice and log probabilities-based performance measurement as the default approach to ensure stability in performance reproduction and minimize unintended interference in evaluating hallucination detection capabilities. To evaluate the multiple-choice in K-HALU, we compute the conditional probabilities of sequence generation, leveraging the auto-regressive of LLMs (Gao et al., 2024). For a sequence $x$ and its input source $S$, the sequence generation probability $P(x)$ is calculated as:*
> > > >
> > > > $P(x) = \frac{1}{|x|} \sum_{i=1}^{|x|} \log \mathbb{P}(x_{i} \mid S:x_{<i})$
> > > >
> > > > *Here, $P(x)$ represents the token probability computed by the model, with ":" indicating sequence concatenation. The input source content $S$ consists of the instruction $I$, the textual document $t$, and the publish date $d$, which can be expressed as $S = [I\text{:}t\text{:}d]$. For each statement from the set of 5 choices, we calculate the cumulative log probabilities for its tokens. These log probabilities are then exponentiated to convert them into actual probabilities. The **Top-$N$ answers** with the highest probabilities are selected as the model's predicted answers.*
> > > >
> > > > *Exact Match: To enable LLMs to provide judgments directly, we set up the task to have the models generate binary outputs (“0” or “1”) indicating the validity of each choice and evaluate them using exact match. We incorporate a post-processing step to minimize errors caused by unnecessary special characters or slight variations in format during the generation process. e.g., “[1, 1, 0, 0, 0]”.*
> > > >
> > > > *LLM-as-a-Judge: We use an LLM-as-a-Judge style prompt to assign scores for hallucination detection results, enabling the LLM to evaluate the quality and accuracy of generated text (Liu et al., 2023; Chiang & Lee, 2023; Zheng et al., 2024). LLMs are tasked with directly generating faithful or hallucinated statements based on the given instructions. These outputs tend to be descriptive, which increases the potential for errors when evaluated using an exact match. To mitigate this issue, we utilize GPT-4 omni (gpt-4o-2024-08-06) as an evaluator to classify the validity of the generated statements as binary values (0 for invalid, 1 for valid)*

---

> > > > > ### Author Response · Authors · 2024-11-25
> > > > > **Plan to improve the script - part 2**
> > > > >
> > > > > > - Regarding the K-HALU samples, we clarified that their source data is publicly accessible and includes sufficient metadata (e.g., publication dates and publishers) to **enable cross-verification**, provided users comply with the terms of use.
> > > > >
> > > > > &rarr; **Improvement**: The source dataset referenced in Footnote 2 is created through funding and planning by government agencies and does not include original links but does contain key metadata that enables cross-verification. Therefore, we will update the K-HALU dataset to include not only the publish date but also the publishers and indices corresponding to the original data.
> > > > >
> > > > > > - We justified the use of **multiple-choice and log probabilities** as evaluation methods by referencing **recent studies, including the Open LLM leaderboard and lm-eval-harness**, to substantiate the validity and robustness of our approach.
> > > > >
> > > > > &rarr; **Improvement**: In **4.1 SETUP**, we have revised the content to explicitly clarify that K-HALU's multiple-choice evaluation is based on log probabilities and added an explanation for why this method was chosen as the default approach. Additionally, we have provided a detailed rationale for this choice in our response to **R4** to address the reviewer's question comprehensively.
> > > > >
> > > > > **[Revision (Page 7, Lines 327 to 329)]**
> > > > >
> > > > > *Log probabilities:
> > > > > We adopt multiple choice and log probabilities-based performance measurement as the default approach to ensure stability in performance reproduction and minimize unintended interference in evaluating hallucination detection capabilities.*
> > > > >
> > > > >
> > > > > > - To address your concerns about evaluation, we added experiments for **Exact Match and LLM-as-a-Judge** methods in **4.1 Setup** and **4.2 Results**.
> > > > >
> > > > > &rarr; **Improvement**: We have enhanced the diversity of analysis and the reliability of results for hallucination detection by incorporating generative evaluation methods, such as Exact Match and LLM-as-a-Judge, into the log probabilities-based approach. Through these generative evaluations, we have cross-verified that open-source LLMs face significant challenges in handling hallucination detection tasks and demonstrated their substantial limitations in instruction-following capabilities.
> > > > >
> > > > > **[Revision (Page 10, Table 8, Lines 475 to 509)]**
> > > > > | Models                        | Exact Match         |                         |                         | LLM-as-a-Judge        |                         |                         |
> > > > > |-------------------------------|---------------------|-------------------------|-------------------------|-----------------------|-------------------------|-------------------------|
> > > > > |                               | Zero-shot          | 3-shot                 | 3-shot + CoT           | Zero-shot            | 3-shot                 | 3-shot + CoT           |
> > > > > | Llama2 (Touvron et al., 2023) | 0.0106             | 0.0051                 | 0.0060                 | 0.0101               | 0.0115                 | 0.0189                 |
> > > > > | Llama3 (AI@Meta, 2024)        | 0.1203             | 0.2396                 | 0.1760                 | 0.0484               | 0.0203                 | 0.0212                 |
> > > > > | KULLM3 (Kim et al., 2024)     | 0.0194             | 0.0083                 | 0.0070                 | 0.1134               | 0.1005                 | 0.0797                 |
> > > > > | ExaOne (Research et al., 2024)| 0.0484             | 0.1885                 | 0.1839                 | 0.0922               | 0.1276                 | 0.1023                 |
> > > > > | Mistral-NEMO (MistralAI, 2024)| 0.1240             | 0.2668                 | 0.2770                 | 0.0995               | 0.1138                 | 0.1106                 |

---

> > > > > > ### Author Response · Authors · 2024-11-25
> > > > > > **Plan to improve the script - part 3**
> > > > > >
> > > > > > >- To complement the strict evaluation approach involving multiple answers, we introduced the **F1 score as an additional metric**.
> > > > > >
> > > > > > &rarr; **Improvement**: In evaluating multiple answers, Accuracy serves as a strict metric that does not allow for partial correctness. To provide a more nuanced assessment of LLM performance, we have conducted experiments incorporating F1, Precision, and Recall as additional evaluation metrics. These experiments enabled us to analyze whether LLMs tend to select more incorrect options or overly restrict their choices. Furthermore, we have updated the evaluation script to measure F1, Precision, and Recall alongside Accuracy, thereby enhancing the insights provided by the K-HALU benchmark.
> > > > > >
> > > > > > **[Revision (Page 17, Table 9, Appendix B F1 Score, Line 842 to 856)]**
> > > > > > | Model                          | Accuracy | F1     | Precision | Recall  |
> > > > > > |--------------------------------|----------|--------|-----------|---------|
> > > > > > | Llama2 (Touvron et al., 2023)  | 0.2475   | 0.3939 | 0.4152    | 0.3979  |
> > > > > > | Llama3 (AI@Meta, 2024)         | 0.2880   | 0.4332 | 0.4503    | 0.4356  |
> > > > > > | KULLM3 (Kim et al., 2024)      | 0.2862   | 0.4156 | 0.4323    | 0.4174  |
> > > > > > | ExaOne (Research et al., 2024) | 0.3295   | 0.4770 | 0.5117    | 0.4760  |
> > > > > > | Mistral-NEMO (MistralAI, 2024) | 0.3014   | 0.4393 | 0.4530    | 0.4417  |
> > > > > > | GPT-3.5 Turbo (Ouyang et al., 2022) | 0.4046 | 0.7237 | 0.6476    | 0.8260  |
> > > > > > | GPT-4 Turbo (OpenAI, 2023)     | 0.7346   | 0.8538 | 0.8211    | 0.8899  |
> > > > > > | GPT-4 omni (OpenAI, 2023)      | 0.7866   | 0.8732 | 0.8552    | 0.8924  |
> > > > > >
> > > > > > > - To assess the feasibility of extending K-HALU, we conducted additional experiments on **English (a high-resource language)** and **Malay (a low-resource language)**, which are now included in **Appendix C**.
> > > > > >
> > > > > > &rarr; **Improvement**: While K-HALU is a dataset designed for hallucination detection using textual sources in the Korean-speaking world, we have enabled cross-linguistic performance comparison by translating the dataset into other languages. For this purpose, we have selected English, the highest-resource language, and Malay, a relatively low-resource language. The experimental results indicate that LLM performance is influenced to some extent by language-specific embedded knowledge and that LLMs continue to face significant challenges in hallucination detection, even when the data is expressed in different languages.
> > > > > >
> > > > > > **[Revision (Page 17, Table 10, Appendix C Extending K-HALU to Multilingual, Line 860 to 912)]**
> > > > > >
> > > > > > **K-HALU (English ver.)**
> > > > > > | Model          | Accuracy | F1     | Precision | Recall  |
> > > > > > |----------------|----------|--------|-----------|---------|
> > > > > > | LLAMA2         | 0.2757   | 0.4124 | 0.4261    | 0.4131  |
> > > > > > | LLAMA3         | 0.2727   | 0.4012 | 0.4139    | 0.4038  |
> > > > > > | KULLM3         | 0.2846   | 0.4082 | 0.4235    | 0.4091  |
> > > > > > | ExaOne         | 0.2638   | 0.4079 | 0.4288    | 0.4106  |
> > > > > > | Mistral NEMO   | 0.2906   | 0.4354 | 0.4496    | 0.4364  |
> > > > > >
> > > > > > **K-HALU (Malay ver.)**
> > > > > > | Model          | Accuracy | F1     | Precision | Recall  |
> > > > > > |----------------|----------|--------|-----------|---------|
> > > > > > | LLAMA2         | 0.2620   | 0.4317 | 0.4393    | 0.4314  |
> > > > > > | LLAMA3         | 0.2715   | 0.4443 | 0.4528    | 0.4452  |
> > > > > > | KULLM3         | 0.2600   | 0.4268 | 0.4373    | 0.4274  |
> > > > > > | ExaOne         | 0.2505   | 0.4171 | 0.4297    | 0.4189  |
> > > > > > | Mistral NEMO   | 0.2772   | 0.4491 | 0.4591    | 0.4476  |
> > > > > >
> > > > > > We sincerely thank you once again for dedicating your time to the review process.
> > > > > >
> > > > > > **A revised version of the manuscript, incorporating the above changes, has been uploaded as a PDF.**

---

> > > > > > > ### Comment · Reviewer_zS85 · 2024-11-27
> > > > > > > **Reviewer Response: Score Adjustment**
> > > > > > >
> > > > > > > I have raised the score for the paper. Thank you for your clarifications, and all the best!

---

> > > > > > > > ### Author Response · Authors · 2024-11-27
> > > > > > > > **Thanks to the Reviewer zS85**
> > > > > > > >
> > > > > > > > Your review has been incredibly helpful in improving our manuscript. Thank you once again!!

---

### Official Review · Reviewer_Fjh4 · 2024-10-31

**Soundness:** 2
**Presentation:** 2
**Contribution:** 3
**Rating:** 6
**Confidence:** 3

**Summary:**

This paper presents a benchmark for evaluating the performance of LLMs on determining whether a statement (short sentence) written in Korean is hallucination or not with respect to a given document also written in Korean and its publication date. Concretely, there is a set of statements for each document and the LLM is tasked to either pinpoint all hallucination ones or all correct ones (i.e., two types of multiple-choice questions). The motivation to create this benchmark is due to the lack of evaluation resources in Korean. The benchmark creation pipeline is described together with its statistics. The performance of several open- and closed-source, multi-lingual and Korean-tuned, LLMs is evaluated.

First, the authors should revise the terminology (i.e., hallucination, factual, faithful) which seems confusing as used in the paper. As the benchmark and task is stated, it seems that the actual task is to distinguish between faithful (i.e., supported by the input document) or unfaithful (not supported by the input document) statements. A statement can be not supported by the given document but still be factual (see definition in Maynez et al. 2020).

Second, it is not clear why the format of the benchmark is multiple-choice and why the strict evaluation on sets of statements (select n hallucination/non-hallucination statements from given N). It is not clear why this task setup, it seems adding extra complexity that is not needed to see whether an LLM can distinguish whether a statement is faithful (or not). Cannot it just evaluate the LLM on each individual statements?

As for the LLM execution of the task, why the selection between hallucination or not is based on probabilities? Why not prompting the model to generate/select the options (i.e., to generate "1, 2")? The authors should explain why they chose this method and may be also add results with the prompt approach. Previous evaluation methods based on prompt ask the model to choose among plausible responses. Also, how was this probability approach implemented for closed-source LLMs (i.e., GPT variants)?

More details about the Chain-of-Thought implementation should be provided. The prompts and partition into reasoning sub-steps?

The authors should avoid using the term "uncertain" throughout the paper. It is confusing and under-specified, do the authors mean that the statement is not supported by the document? Then should be said 'unsupported'.

Related work should discuss other work on creating evaluation resources for less represented languages. eg., [1].
[1] https://arxiv.org/abs/2403.20266

Section qualitative analysis, Appendix A and Figure 4 should go into the main paper.

Minor comments:

- the source dataset should be better described (Knowledge Graph-to-Text dataset from AI-Hub) what is this graph?
- Lens Observation section. At this point in the paper, it is not clear what the used frameworks are nor why suddenly we are looking at these frameworks.
- Line 360, "handling hallucinations" should be called "detecting hallucinations" as the task is to detect whether the sentence is hallucination or not rather than doing something to mitigate hallucinations.

**Strengths:**

- The collected evaluation suit from Korean sources is valuable for the evaluation of Korean language models.
- Experiments show that LLMs struggle to reason about content in Korean (though see comments above about the experimental setup).

**Weaknesses:**

Execution of the work: task formulation (the formulation of the task on multiple-choice seems unjustified and unnecessary more complex) and implementation with LLMs only using probabilities.

Overall, the contribution seems rather thin.

**Questions:**

- why the format of the benchmark?
- why only the implementation using probabilities?

**Details Of Ethics Concerns:**

To create the benchmark the authors mention that the sources are publicly available, may be they should state the terms of use or licence?

---

> ### Author Response · Authors · 2024-11-21
> **Response to Reviewer Fjh4 - Part 1**
>
> We sincerely thank Reviewer Fjh4 for the review and are grateful for the time you spent with our submission. Here we answer all the questions you posted.
>
> **Q1. Authors should revise the terminology.**
>
> **R1.** The issue regarding the terms hallucinated statements and factual statements is considered to have arisen because of the incomplete mapping of the English terms "faithfulness" and "unfaithfulness" to the Korean terms. Specifically, for the Korean dataset and prompt settings, the terms hallucinated (Kor: 환각) and factual (Kor: 사실) are selected as more appropriate terminology that does not distort the contextual intent.
>
> Moreover, the well-known hallucination benchmark dataset HaluEVAL [1] also uses the terms hallucinated statements and factual statements when assessing whether an input sentence contains hallucination.
>
> However, we agree with the reviewer's concern that some sentences expressed as factuality or factual statements might lead to confusion among readers based on recent trends in the terminology definitions used in hallucination research. To address this, **we have revised the terminology as follows**.
>
> (1) *Hallucinated statements / Factual statements*
>
> → **Hallucinated / Faithful statements**
>
> (2) *Factuality*
>
> → **Faithfulness**
>
> [1] Li, J., Cheng, X., Zhao, W. X., Nie, J. Y., & Wen, J. R. (2023, December). HaluEval: A Large-Scale Hallucination Evaluation Benchmark for Large Language Models. In Proceedings of the 2023 Conference on Empirical Methods in Natural Language Processing (pp. 6449-6464).
>
> **Q2. It is not clear why the format of the benchmark is multiple-choice and why the strict evaluation on sets of statements. Cannot it just evaluate the LLM on each individual statements?**
>
> **R2.** We adopt the multiple-choice format to mitigate issues that could arise in generative evaluations, such as length imbalance, variability in sentence expressions, and structural differences that may distort performance. It also reduces numerical instability caused by summing log probabilities and ensures consistent evaluation scores. By minimizing unintended factors in performance assessment, we believe this approach provides a more valid comparison of hallucination detection performance across models.
>
> Moreover, the multiple-choice format has been widely adopted in benchmark datasets, as demonstrated by its use in Huggingface’s OpenLLM Leaderboard (https://huggingface.co/spaces/open-llm-leaderboard-old/open_llm_leaderboard). This method of comparing model performance using log probabilities and multiple-choice questions has been validated through thousands of model evaluations within the NLP community [1,2,3,4]. For instance, TruthfulQA [4], a benchmark dataset highly relevant to hallucination evaluation and included in Huggingface’s H4 leaderboard, officially supports multiple-choice and *select n statements* formats (see [TruthfulQA Multiple Choice](https://huggingface.co/datasets/truthfulqa/truthful_qa/viewer/multiple_choice?row=1)). This demonstrates that both evaluating individual statements and using a multiple-choice format are viable methodologies for constructing hallucination detection benchmarks.
>
> Additionally, we argue that the multiple-choice format, particularly the *select n hallucination statements* setup, serves as an effective means of applying stricter evaluation criteria to benchmark datasets. Generative evaluation methods often struggle to enforce rigorous criteria for multiple correct answers, tending to allow for a broader range of generated outputs. By contrast, our approach explicitly incorporates 40% of the dataset as multi-answer questions, leveraging the inherent difficulty of the multiple-choice format to provide more diverse insights into model performance. This setup not only allows for a more nuanced analysis of models evaluated with the K-HALU benchmark but also capitalizes on the advantages of difficulty settings unique to the multiple-choice approach.
>
> [1] Zellers, R., Holtzman, A., Bisk, Y., Farhadi, A., & Choi, Y. (2019, July). HellaSwag: Can a Machine Really Finish Your Sentence?. In *Proceedings of the 57th Annual Meeting of the Association for Computational Linguistics* (pp. 4791-4800).
>
> [2] Clark, P., Cowhey, I., Etzioni, O., Khot, T., Sabharwal, A., Schoenick, C., & Tafjord, O. (2018). Think you have solved question answering? try arc, the ai2 reasoning challenge. *arXiv preprint arXiv:1803.05457*.
>
> [3] Hendrycks, D., Burns, C., Basart, S., Zou, A., Mazeika, M., Song, D., & Steinhardt, J. Measuring Massive Multitask Language Understanding. In *International Conference on Learning Representations*.
>
> [4] Lin, S., Hilton, J., & Evans, O. (2022, May). TruthfulQA: Measuring How Models Mimic Human Falsehoods. In *Proceedings of the 60th Annual Meeting of the Association for Computational Linguistics (Volume 1: Long Papers)* (pp. 3214-3252).

---

> ### Author Response · Authors · 2024-11-21
> **Response to Reviewer Fjh4 - Part 2**
>
> **Q3. Why the selection between hallucination or not is based on probabilities? Why not prompting the model to generate/select the options? The authors should explain why they chose this method and may be also add results with the prompt approach.**
>
> **R3.** We explained why log probabilities are used for evaluation in **R2**. To further address the reviewer's question, we provide the following clarification:
>
> Evaluating performance based on generated outputs introduces confounding factors related to instruction-following abilities rather than purely assessing hallucination detection. Furthermore, while a model may generate a correct answer as intended, the inherently descriptive nature of LLM-generated outputs can result in mismatches during exact match evaluations, leading to unintended false negatives. (We acknowledge that instruction-following capabilities are a significant aspect of evaluation, as emphasized in benchmarks like MMMLU [1]. However, our primary focus was on the task of hallucination detection.)
>
> To address the reviewer’s concern, **we have included additional experiments using two evaluation methods in 4.1 Setup and 4.2 Results; Exact Match and LLM-as-a-Judge**:
>
> **Exact Match**: To enable LLMs to provide judgments directly, we set up the task to have the models generate binary outputs (“0” or “1”) indicating the validity of each choice and evaluate them using exact match. We incorporate a post-processing step to minimize errors caused by unnecessary special characters or slight variations in format during the generation process. e.g., “[1, 1, 0, 0, 0]”.
>
> **LLM-as-a-Judge**: We use an LLM-as-a-Judge style prompt to assign scores for hallucination detection results, enabling the LLM to evaluate the quality and accuracy of generated text (Liu et al., 2023; Chiang & Lee, 2023; Zheng et al., 2024). LLMs are tasked with directly generating faithful or hallucinated statements based on the given instructions. These outputs tend to be descriptive, which increases the likelihood of errors when evaluated using exact match or overlap-based methods. To mitigate this issue, we utilize GPT-4 omni (gpt-4o-2024-08-06) as an evaluator to classify the validity of the generated statements as binary values (0 for invalid, 1 for valid).
>
> **Exact Match Performance of LLMs**
>
> | Model          | Zero-shot | 3-shot  | 3-shot + CoT |
> |----------------|-----------|---------|--------------|
> | LLAMA2         | 0.0106    | 0.0051  | 0.006        |
> | LLAMA3         | 0.1203    | 0.2396  | 0.176        |
> | KULLM3         | 0.0194    | 0.0083  | 0.007        |
> | ExaOne         | 0.0484    | 0.1885  | 0.1839       |
> | Mistral NEMO   | 0.124     | 0.2668  | 0.277        |
>
> When using exact match to evaluate performance, it becomes more challenging for models to achieve high accuracy compared to multiple-choice accuracy. This is due to the involvement of instruction-following abilities, which highlights more pronounced performance differences among models. Similar to evaluations based on multiple-choice and log probabilities, applying shots and CoT reasoning shows partial performance improvement. However, consistent performance enhancement is not observed across all settings.
>
> **LLM-as-a-Judge Performance of LLMs**
> | Model          | Zero-shot | 3-shot  | 3-shot + CoT |
> |----------------|-----------|---------|--------------|
> | LLAMA2         | 0.0101    | 0.0115  | 0.0189       |
> | LLAMA3         | 0.0484    | 0.0203  | 0.0212       |
> | KULLM3         | 0.1134    | 0.1005  | 0.0797       |
> | ExaOne         | 0.0922    | 0.1276  | 0.1023       |
> | Mistral NEMO   | 0.0995    | 0.1138  | 0.1106       |
>
> To address potential distortions caused by descriptive outputs in exact match evaluations, we used GPT-4o to validate the generated results. However, even in this evaluation setup, baseline open-source LLMs consistently recorded very low performance. Interestingly, KULLM3 performed better in this evaluation compared to other methods. Additionally, ExaOne and Mistral NEMO demonstrated generally higher performance across log probability, exact match, and LLM-as-a-Judge evaluations.
>
> These results highlight the challenges open-source LLMs face in directly solving K-HALU’s hallucination detection tasks through generative outputs or CoT prompting. This implies the need for further research to improve model capabilities in hallucination detection tasks effectively.
>
> [1] https://huggingface.co/datasets/openai/MMMLU
>
> **Q4. how was this probability approach implemented for closed-source LLMs?**
>
> **R4.** We acknowledge the reviewer’s valid observation regarding an oversight in describing the evaluation of closed API models. As these APIs do not provide log probabilities, we instead used the generated results (e.g., hallucinated statement indices) and evaluated them based on exact match criteria. **This clarification has been explicitly updated in Lines 320-325 and Footnote 4**.
>
> Thank you for pointing this out.

---

> ### Author Response · Authors · 2024-11-21
> **Response to Reviewer Fjh4 - Part 3**
>
> **Q5. More details about the Chain-of-Thought implementation should be provided. The prompts and partition into reasoning sub-steps?**
>
> **R5.** The CoT prompts are designed to explain which part of the input document or statement contains errors, leading to the determination of hallucination. The prompts include three examples illustrating the reasoning process.
>
> For all three evaluation methods, **we have included the CoT prompts in Appendix G**, along with supplementary explanations. Below is an example of a CoT prompt. For brevity, considering the length constraints of OpenReview, the example has been reduced from three to one and includes a condensed version of the document content:
>
> **[CoT Prompt Example of Exact Match]**
>
> *Document Date: January 11, 2021*
>
> *Document Content: The Korean Association for Public Administration (KAPA) ... and then attended the live broadcast of former President Roh Moo-hyun’s inauguration the next day.*
>
> *Identify Hallucinated Sentences in the Document*
>
> *Options:*
>
> *1. Professor Soon-Ae Park strives to expand women’s participation in the public sector.*
>
> *Analysis: The document indirectly suggests efforts related to women but doesn’t explicitly state this.*
>
> *Conclusion: Not hallucinated → 0*
>
> ...
>
> *4. Professor Soon-Ae Park said she aims to change the administrative field, burdened by the heavy armor of the 3G era.*
>
> *Analysis: The document refers to the "2G era," not the "3G era."*
>
> *Conclusion: Incorrect information → 1*
>
> *5. Professor Soon-Ae Park believes civil servants prefer maintaining regulations over abolishing them.*
>
> *Analysis: While this is indirectly implied through comparisons with the U.S., it can be inferred as true.*
>
> *Conclusion: Factually supported → 0*
>
> *Answer: [0,0,1,1,0]*
>
> **Q6. Authors should avoid using the term "uncertain" to  "unsupported."**
>
> **R6.** We appreciate the reviewer’s feedback on terminology. To improve clarity, we have replaced the term "uncertain" with "unsupported."
>
> **Q7. Related work should discuss other work on creating evaluation resources for less represented languages. eg., [1] https://arxiv.org/abs/2403.20266.**
>
> **R7.** We appreciate the reviewer's suggestion and will strengthen the motivation for K-HALU as an evaluation resource by incorporating the proposed references into the Introduction. This emphasizes the unique challenges in creating and utilizing hallucination benchmarks for non-English languages like Korean and further supports the motivation behind developing K-HALU.
>
> **Revised (Lines 50-79):**
> *Furthermore, most benchmarks focus on parametric knowledge and linguistic nuances specific to the English-speaking world, making them less ideal as resources for evaluating underrepresented languages [1]. In particular, English hallucination benchmark datasets are also challenging to apply to the Korean language caused by linguistic and socio-cultural differences (Hendrycks et al., 2021; Seo et al., 2024).*
>
> [1] Julen Etxaniz, Oscar Sainz, Naiara Miguel, Itziar Aldabe, German Rigau, Eneko Agirre, Aitor Ormazabal, Mikel Artetxe, and Aitor Soroa. 2024. Latxa: An Open Language Model and Evaluation Suite for Basque. In Proceedings of the 62nd Annual Meeting of the Association for Computational Linguistics (Volume 1: Long Papers), pages 14952–14972, Bangkok, Thailand. Association for Computational Linguistics.
>
> **Q8. Appendix A and Figure 4 should go into the main paper.**
>
> **R8.** In response to the reviewer’s suggestion, we have relocated the section on **Lens Observation to Appendix D**.
>
> **The descriptions of Figure 4 from Appendix A: Qualitative Details have been moved to Lines 512–514 in the main body**. However, we hope the reviewer understands that spatial constraints prevent us from moving Figure 4 itself into the main text.
>
> **Q9. The source dataset should be better described.**
>
> **R9.** **We have revised the 3.2 Dataset Creation section to include the following description about the Source Dataset starting from Lines 205-209**:
>
> *This dataset consists of textual documents, including news articles, magazine articles, and books, describing faithful relationships in the form of statements. Each statement is tagged as subject, predicate, and object, forming a triples-context pair. Each document contains one or more triples-context pairs, amounting to a total of 300,178 samples, each of which has processed human labeling and data refinement.*
>
> **Q10. "handling hallucinations" should be called "detecting hallucinations"**
>
> **R10.** Thank you for pointing this out. We agree with the reviewer’s suggestion and have revised the phrasing for clarity and accuracy. Specifically, we replaced "handling hallucinations" with "detecting hallucinations" in **lines 82 and 364** to better align with the task's focus on determining whether a sentence is a hallucination rather than mitigating it. This change will be included in the updated manuscript.

---

> ### Author Response · Authors · 2024-11-21
> **Response to Reviewer Fjh4 - Part 4**
>
> **Q11.Execution of the work: task formulation and implementation with LLMs only using probabilities.**
>
> **R11.** The reason for adopting multiple-choice and log probabilities as our benchmark evaluation methods has already been explained in **R2** and **R3**. Furthermore, we have considered the reviewer’s concerns and incorporated Exact Match and LLM-as-a-Judge to deliver a more comprehensive and balanced analysis.
>
> **Q12. why the format of the benchmark?**
>
> **R12.** As mentioned in lines 80–82 of the manuscript, there is a lack of publicly available Korean NLP hallucination benchmark datasets for evaluating open-source models. Therefore, we design an evaluation method that allows anyone to easily measure the hallucination detection capabilities of open-source LLMs. We believe this approach is more realistic and sustainable over time, as it considers scenarios where multiple statements, including hallucinated ones, coexist rather than limiting the task to binary classification for individual samples.
>
> Additionally, generative evaluation methods increase the complexity of assessing model performance, as factors such as instruction-following abilities and post-processing can influence the scores. The LLM-as-a-Judge approach, while insightful, comes with high costs and challenges in full reproducibility. Moreover, we leverage bootstrap evaluation provided by lm-eval-harness (https://github.com/EleutherAI/lm-evaluation-harness) for log probabilities and the multiple-choice format. By repeatedly sampling and evaluating multiple times, we ensure the reliability and robustness of the model evaluations.
>
> **Q13. why only the implementation using probabilities?**
>
> **R13.** Using log probabilities helps mitigate issues that can arise in generative evaluations, such as length imbalances, variability in sentence expressions, and interference from structural factors. By relying on the summation of log probabilities, this method reduces numerical instability and ensures relatively consistent evaluation scores. This approach minimizes unintended factors in performance assessment, making it more reliable for comparing hallucination detection capabilities across models.
>
> As noted in previous responses, many benchmark datasets adopt log-probability-based evaluation due to its robustness against variations in performance during reimplementation and its cost-effectiveness for evaluation. Additionally, as demonstrated in the additional experiments in **R3**, the results of Exact Match and LLM-as-a-Judge align closely with those obtained using log probabilities, showing consistent results.
>
> We agree that including an analysis of generative evaluation methods alongside log-probability-based results would strengthen the paper. Therefore, we will incorporate the experimental findings from **R3** into the manuscript to provide a more comprehensive and balanced discussion.
>
> **License**
>
> **We have included a section in Appendix E** to address the licensing guidelines associated with the use of AI-HUB Open data. The following details will be outlined:
>
> **[AI-HUB License]**
> *The AI Open dataset provided by AI-HUB (referred to as "AI Data") was developed as part of the "Intelligent Information Industry Infrastructure Construction" project supported by the Ministry of Science and ICT and the National Information Society Agency of Korea (NIA). Below are the key guidelines for its use:*
>
> *All rights to the AI Data, including the data, AI application models, source code for data authoring tools, and manuals, belong to the performing and participating institutions involved in the construction of the data and the NIA.*
>
> *The AI Data was created to advance AI technologies, products, and services. It can be used for commercial and non-commercial research and development purposes across various fields, including intelligent products and services, chatbots, and more.*
>
> *Any use of AI Data must acknowledge it as a result of the NIA's projects. This attribution must also apply to derivative works.*
>
> To comply with these guidelines, the following acknowledgment will be added to the manuscript:
>
> *This research (paper) used datasets from 'The Open AI Dataset Project (AI-Hub, S. Korea)'. All data information can be accessed through 'AI-Hub (www.aihub.or.kr)'.*
>
> Additionally, the benchmark dataset derived from the AI-HUB data will be made publicly available as "external data" upon agreement with the NIA. This process aligns with the approach used in prior Korean benchmark [1] based on AI-HUB data, as demonstrated by:
>
> [1] Seo, J., Lee, S., Park, C., Jang, Y., Moon, H., Eo, S., ... & Lim, H. S. (2022, July). A dog is passing over the jet? A text-generation dataset for Korean commonsense reasoning and evaluation. In Findings of the Association for Computational Linguistics: NAACL 2022 (pp. 2233-2249).
>
> We sincerely thank Reviewer Fjh4 for the thoughtful review and valuable feedback. We hope our detailed responses have effectively addressed your concerns.

---

> > ### Author Response · Authors · 2024-11-25
> > **Looking forward to your response**
> >
> > Dear Reviewer Fjh4,
> >
> > Thank you for taking the time to review our work and provide valuable and constructive feedback. A few days ago, **we submitted our revised manuscript along with our responses to your comments**. We would greatly appreciate it if you could share any additional thoughts or concerns so we can address them and continue to refine our work.
> >
> > In our previous response, we made the following revisions to address the concerns you raised:
> >
> > - To enhance clarity and reduce ambiguity, we revised key terms: *uncertain* was replaced with **unsupported**, *handling hallucinations* was changed to **detecting hallucinations**, and *factuality/factual* was updated to **faithfulness/faithful**.
> >
> > - **Updated the discussion of related work in 1. Introduction, Lines 50-79**, incorporating the papers you suggested.
> >
> > - Enhanced the explanation of the source dataset in **3.2 Dataset Creation, Lines 205-209**.
> >
> > - Provided a rationale for using **multiple-choice and log probabilities** as evaluation methods, **referencing recent studies such as the Open LLM leaderboard and lm-eval-harness to substantiate the validity and effectiveness of our approach**.
> >
> > - Addressed your concerns regarding evaluation by adding experiments for **Exact Match and LLM-as-a-Judge methods in 4.1 Setup and 4.2 Results**.
> >
> > - Moved the explanation of **lens observations to Appendix D** and **added further details to the qualitative analysis**.
> >
> > - **Included specific prompts used for CoT reasoning in Appendix G** to clarify the use of log probabilities, exact match, and LLM-as-a-Judge approaches.
> >
> > - Corrected errors in the explanation of closed-API evaluation methods in **4.1 Setup, Lines 320-325**.
> >
> > - Added details regarding the **AI-HUB license in Appendix E**.
> >
> > We have diligently addressed almost all your concerns and suggestions in the revised manuscript. We sincerely hope these revisions meet your expectations and look forward to your kind and thorough reassessment.
> >
> > Thank you again for your valuable feedback and time.
> >
> > Best regards,
> >
> > The Authors

---

> ### Comment · Reviewer_Fjh4 · 2024-11-27
>
> Thank you for your response. Some points have been clarified. However, the following discussion point has not been entirely clarified to me:
>
> - L196, description of the k-HALU. I still think this formulation is unnecessarily strict. Although the authors partially address this by reporting F1 metrics. However, from your response (quoted below [1]) I'm not sure why the task cannot be defined on pairs such as (document, statement) and then do detection hallucination/faithful on these pairs? That is, instead of having the task to be (document, statement1, statement2, ...., statementN) hallucination/faithful?; having the tasks (document, statement1) hallucination/faithful? + (document, statement2) hallucination/faithful? + , ...., + (document, statementN) hallucination/faithful? as individual test cases? In the end, what we want to know is whether the LLMs are good at reasoning about each of the statements. Having to reason about all of them at the same time can be a harder task (or not?). Perhaps, the authors should include
>
> [1] "We adopt the multiple-choice format to mitigate issues that could arise in generative evaluations, such as length imbalance, variability in sentence expressions, and structural differences that may distort performance. It also reduces numerical instability caused by summing log probabilities and ensures consistent evaluation scores. By minimizing unintended factors in performance assessment, we believe this approach provides a more valid comparison of hallucination detection performance across models."
>
>
> - Other minor comments from re-reading the pdf.
>
> L176, "K-HALU is a hallucination benchmark composed of 2,170 multiple-answer questions" ==> this description is a bit misleading, this is not a QA dataset.
>
> L188, "and their structure varies according to the instruction type and the number of correct answer" ==> what you mean by 'structure'?
>
> Figure 1, still contains the term "uncertain"
>
> Table 4, caption should describe the metric reported in the Table's cells.

---

> ### Author Response · Authors · 2024-11-27
> **Response to Reviewer Fjh4 - Part 2.1**
>
> We sincerely thank the reviewer for taking the time to review our revised manuscript and responses. We greatly appreciate your efforts and thoughtful feedback. To address any remaining concerns, we have made further revisions to the manuscript, as detailed in the responses below. Our responses are as follows:
>
> **Q1.**: L196, description of the k-HALU. I still think this formulation is unnecessarily strict. I'm not sure why the task cannot be defined on pairs such as (document, statement) and then do detection hallucination/faithful on these pairs? That is, instead of having the task to be (document, statement1, statement2, ...., statementN) hallucination/faithful?; having the tasks (document, statement1) hallucination/faithful? + (document, statement2) hallucination/faithful? + , ...., + (document, statementN) hallucination/faithful? as individual test cases? In the end, what we want to know is whether the LLMs are good at reasoning about each of the statements. Having to reason about all of them at the same time can be a harder task (or not?).
>
> **A1.**: Thank you for your thoughtful feedback. We understand your concern regarding the potential complexity of reasoning about all statements simultaneously. As described in the dataset examples, each sample consists of a single document accompanied by five choices. In the Exact Match and LLM-as-a-Judge evaluation methods, the model generates the correct answers by considering all statements within a single document at once. We believe that generating multiple answers in these evaluation methods reflects a more real-world and comprehensive approach to detecting hallucinations by accounting for multiple contributing factors.
>
> However, a log probability-based multiple choice task evaluates each statement independently in conjunction with the document. This effectively forms pairs such as (document, statement1), (document, statement2), ..., (document, statement5), enabling the independent evaluation of each statement for hallucination/faithfulness detection. In this approach, the model calculates the log probabilities of the five statements independently and selects the Top-N answers based on these probabilities.
>
> Therefore, the log probability-based multiple choice task in K-HALU evaluates whether the model can make valid inferences for multiple answers without being disrupted by other statements. It strictly evaluates whether valid reasoning has been applied to choices that were not considered in single-answer tasks. By incorporating F1 scoring for partial correctness, this approach provides deeper insights into the evaluation targets.
>
> K-HALU adopts multiple answers to enable more realistic and discriminative evaluations and offers results that consider statements evaluated both collectively and individually. We kindly hope the reviewer regards our benchmark’s capability to provide such comprehensive and high-resolution evaluations.
>
> Also, we acknowledge that the insufficient explanation of "For any sequence" and $x$ in L331 may have caused ambiguity in describing the evaluation approach. To address this issue, we will incorporate the details from [1] to clarify the evaluation method.
>
> **[L327]**
>
> "**Log probabilities**:
> We adopt multiple choice and log probabilities-based performance measurement as the default approach to ensure stability in performance reproduction and minimize unintended interference in evaluating hallucination detection capabilities. To evaluate the multiple-choice in K-HALU, we compute the conditional probabilities of sequence generation, leveraging the auto-regressive of LLMs (Gao et al., 2024). **For each statement (choice) $x$ and its input source $S$, the sequence generation probability $P(x)$ is calculated as**:
>
> $P(x) = \frac{1}{|x|} \sum_{i=1}^{|x|} \log \mathbb{P}(x_{i} \mid S:x_{<i})$
>
> Here, $P(x)$ represents the token probability computed by the model, with ":" indicating sequence concatenation. The input source content $S$ consists of the instruction $I$, the textual document $t$, and the publish date $d$, which can be expressed as $S = [I\text{:}t\text{:}d]$.
>
> **For each of the 5 choices, the cumulative log probabilities of tokens are calculated independently by concatenating the input source $S$ with each choice $x$. Finally, the Top-N answers, corresponding to the number of correct answers for the task, are selected based on their probabilities. This approach minimizes unintended interference from other choices and mitigates score distortion caused by differences in choice length and structure, maintaining stability in performance reproduction.**"
>
> Furthermore, we sincerely appreciate the reviewer’s suggestion, which led us to incorporate evaluations using both the Exact Match and LLM-as-a-Judge methods. These additions allow us to compare the model's performance on statements both collectively and independently, thereby enhancing the diversity of evaluation methods in the manuscript and strengthening the results.

---

> ### Author Response · Authors · 2024-11-27
> **Response to Reviewer Fjh4 - Part 2.2**
>
> **Q2.** L176, "K-HALU is a hallucination benchmark composed of 2,170 multiple-answer questions" ==> this description is a bit misleading, this is not a QA dataset.
>
> **A2.** Thank you for your thoughtful feedback. The section has been revised as follows:
>
> **[L176]**
>
> "K-HALU is a hallucination detection benchmark composed of 2,170 multiple-choice tasks, where there can be more than one correct answer."
>
> **Q3.** L188, "and their structure varies according to the instruction type and the number of correct answer" ==> what you mean by 'structure'?
>
> **A3.** Our intent with this sentence was to convey that the proportion of hallucinated or faithful statements varies across the five statements in each document. We agree that the original phrasing was ambiguous and have revised it as follows:
>
> **[L188]**
>
> "The statements are categorized as either hallucinated or faithful, and the proportion of these categories varies depending on the instruction type and the number of correct answers."
>
> **Q4.** Figure 1, still contains the term "uncertain"
>
> **A4.** Thank you for pointing out the overlooked issue during the revision process. The part mentioned by the reviewer is the English translation of a Korean prompt, so a simple adjustment in expression should suffice. The term **"uncertain"** in **L61** have been revised to **"unsupported"**.
>
> **Q5.** Table 4, caption should describe the metric reported in the Table's cells.
>
> **A5.** The caption of Table 4 has been revised to clarify that accuracy was evaluated using log probabilities.
>
> **[Table 4 caption]**
>
> "Model performance across seven knowledge domains, with accuracy evaluated using log probabilities. Bold indicates the domain where each model achieved the highest performance, while underline represents the domain where each model recorded the lowest performance."
>
> Additionally, we have revised the captions for **Tables 5, 6, and 7** to clearly indicate the metrics used.
>
> Thank you for taking the time to review our work and provide your valuable insights.

---

> > ### Author Response · Authors · 2024-11-28
> > **Manuscript Update**
> >
> > We have followed the helpful suggestions from the reviewer and updated the manuscript accordingly.
> >
> > Once again, we thank you for your valuable feedback, which has helped strengthen our paper. We are happy to continue the discussion if you have any further questions.

---

> ### Comment · Reviewer_Fjh4 · 2024-12-02
>
> Thank you for the detailed response. The authors should better motivate the choice of formulating the task as a multiple choice. To say that "is more realistic" is not enough and too general; I still struggle to find or see the situation why we want to evaluate several statements at the same time. Maybe now that the authors also incorporate individual evaluation, altogether, it is just different approaches to evaluate. Because the corpus can be a valuable resource I increased my score.

---

> ### Author Response · Authors · 2024-12-02
> **Thanks to the Reviewer Fjh4**
>
> Thanks a lot for your thorough reviews and for recognizing the value of our proposed resource, which has resulted in raising the scores. We're truly glad to hear that our revisions have strengthened the paper and helped address your concerns.
>
> To resolve the remaining questions raised by the reviewer, we will revise the manuscript as follows:
>
>
> (1) To provide a detailed explanation of the rationale behind the multiple choice format, we will add a new section in the Appendix.
>
> **[Appendix: Multiple Choice Task]**
>
> We adopt the multiple choice format as the default evaluation method to mitigate issues that could arise in generative evaluations, such as length imbalance, variability in sentence expressions, and structural differences that may distort performance. By minimizing unintended factors in performance assessment, we believe this approach ensures a more valid comparison of hallucination detection performance across models.
>
> Additionally, we address potential numerical instability by summing the log probabilities of each token in a choice and normalizing by the length of the choice. We also leverage bootstrap evaluation provided by the lm-eval-harness tool\footnote{\url{https://github.com/EleutherAI/lm-evaluation-harness}} for log probabilities and the multiple choice format. By repeatedly sampling and evaluating, this method ensures consistently reproducible scores and improves the reliability of model evaluations.
>
> In the NLP community, the multiple choice format has been widely adopted in benchmark datasets, as demonstrated by its use in Huggingface’s OpenLLM Leaderboard\footnote{\url{https://huggingface.co/spaces/open-llm-leaderboard-old/open_llm_leaderboard}}. This method of comparing model performance using log probabilities and multiple choice questions have been validated through thousands of model evaluations [1,2,3,4]. For instance, TruthfulQA [4], a benchmark dataset highly relevant to hallucination evaluation and included in Huggingface’s H4 leaderboard, officially supports multiple-choice and select-n-statements formats. This highlights that both evaluating multiple statements and using a multiple choice format are well-validated and effective approaches for constructing hallucination detection benchmarks.
>
> [1] Zellers, R., Holtzman, A., Bisk, Y., Farhadi, A., & Choi, Y. (2019, July). HellaSwag: Can a Machine Really Finish Your Sentence?. In Proceedings of the 57th Annual Meeting of the Association for Computational Linguistics (pp. 4791-4800).
>
> [2] Clark, P., Cowhey, I., Etzioni, O., Khot, T., Sabharwal, A., Schoenick, C., & Tafjord, O. (2018). Think you have solved question answering? try arc, the ai2 reasoning challenge. arXiv preprint arXiv:1803.05457.
>
> [3] Hendrycks, D., Burns, C., Basart, S., Zou, A., Mazeika, M., Song, D., & Steinhardt, J. Measuring Massive Multitask Language Understanding. In International Conference on Learning Representations.
>
> [4] Lin, S., Hilton, J., & Evans, O. (2022, May). TruthfulQA: Measuring How Models Mimic Human Falsehoods. In Proceedings of the 60th Annual Meeting of the Association for Computational Linguistics (Volume 1: Long Papers) (pp. 3214-3252).
>
> (2) To explain the advantages of the multiple answer evaluation method, we will revise the content in **L340** as follows:
>
> **L340**
>
> The multiple-answer evaluation method enables LLMs to independently assess the faithfulness of each statement, allowing for a more detailed and nuanced evaluation. This approach mitigates issues where answers are determined based on the most plausible option or superficial similarity rather than a complex understanding of the relationship between the document and the statement [1,2].
>
> [1] Khashabi, D., Chaturvedi, S., Roth, M., Upadhyay, S., & Roth, D. (2018). Looking Beyond the Surface: A Challenge Set for Reading Comprehension over Multiple Sentences. In Proceedings of NAACL-HLT (pp. 252-262).
>
> [2] Lin, S., Hilton, J., & Evans, O. (2022, May). TruthfulQA: Measuring How Models Mimic Human Falsehoods. In Proceedings of the 60th Annual Meeting of the Association for Computational Linguistics (Volume 1: Long Papers) (pp. 3214-3252).
>
> Thank you once again for taking the time to review our work. We sincerely appreciate your generous reconsideration.

---

### Official Review · Reviewer_cEAG · 2024-11-06

**Soundness:** 2
**Presentation:** 3
**Contribution:** 2
**Rating:** 6
**Confidence:** 3

**Summary:**

This work introduced a Korean benchmark namely K-HALU designed to evaluate LLM's hallucination detection in Korean. Specifically, K-HALU consists of more than 2,000 challenging test samples from Korean news, magazines and books, covering seven domains. The evaluated LLMs involve open-source models: Llama 2, Llama3, Mistral and Korean models such as KULLM3 and ExaOne, and API models: GPT-3.5 and GPT-4 series. Results show that the open-source LLMs exhibit low accuracy less than 35% and 15% on harder tasks, compared with the API LLMs show better accuracy by 27%.

**Strengths:**

1. This paper introduces a Korean benchmark to evaluate LLM's ability for hallucination detection. The benchmark consists of more than 2,000 examples and is labeled by both human annotators and top-performing models.

2. The writing is clear and easy to follow.

3. The analyses are extensive for different LLMs and different scenarios such as domain, instruction type and few-shot.

**Weaknesses:**

1. The comparisons and discussion with the English are missing. Given the fact that there exist benchmarks for hallucination detection in English, the difficulty of Korean hallucination detection by LLMs and the corresponding novelty is unclear to me. This will hurt the contribution of this work.

2. The evaluation measure is a bit simple to some extent. The measurement used in the benchmark is selecting the hallucinated statements from the given statements. However, it is more useful for LLMs to directly give judgments or reasons on whether an input has a hallucination. The problem setting in such two cases is different in the existence of reference selections.

3. The metrics on how different human annotators consensus with each other is unclear. Besides, the linguistic background of human experts can be essential to the correctness of data labeling.

4. (Minor) I notice from Table 7 that The CoT approach can hardly improve the accuracy, it is peculiar from the literature of CoT. More discussion on this can be helpful.

5. (Minor) The question in the benchmark can have multiple selections, thus other measurements such as precision, recall or F1 can be more comprehensive to evaluate hallucination detection than accuracy.

**Questions:**

I hope the authors can clarify my concerns about the above weakness. But if I missed some important thing, please correct me in the rebuttal.

---

> ### Author Response · Authors · 2024-11-21
> **Response to Reviewer cEAG - Part 1**
>
> We sincerely thank reviewer cEAG for the valuable comments. Here we answer all the questions you posted.
>
> **Q1. The comparisons and discussion with the English are missing.**
>
> **R1**. Most hallucination benchmarks in English focus on hallucinations related to knowledge specific to the English-speaking world. Evaluating hallucinations in English, both in input documents and embedded knowledge, has been extensively studied, and the hallucination performance of the open-source LLMs has already been reasonably well-validated in English.
>
> However, **K-HALU** uses Korean textual documents as the knowledge source, making it a benchmark specifically designed for Korean, which inherently incorporates socio-cultural differences that English benchmarks cannot account for (lines 51-52, 78–79). For example, topics such as the Korean financial system, the history of Goguryeo, or relations with North Korea are crucial knowledge sources for assessing the reliability of LLMs in the Korean NLP community. However, such topics are challenging to consider through existing English benchmarks.
>
> We acknowledge the reviewer’s point and have taken it into consideration. To address your concern, we have extended the dataset by translating it into a high-resource language (English) and a low-resource language (Malay) and evaluated the corresponding performances. These results and settings have been included in the **Appendix C Extending K-HALU to Multilingual**
>
> **Performance of Models on K-HALU (English and Malay Versions)**
>
> **K-HALU (English ver.)**
> | Model          | Accuracy | F1     | Precision | Recall  |
> |----------------|----------|--------|-----------|---------|
> | LLAMA2         | 0.2757   | 0.4124 | 0.4261    | 0.4131  |
> | LLAMA3         | 0.2727   | 0.4012 | 0.4139    | 0.4038  |
> | KULLM3         | 0.2846   | 0.4082 | 0.4235    | 0.4091  |
> | ExaOne         | 0.2638   | 0.4079 | 0.4288    | 0.4106  |
> | Mistral NEMO   | 0.2906   | 0.4354 | 0.4496    | 0.4364  |
>
> **K-HALU (Malay ver.)**
> | Model          | Accuracy | F1     | Precision | Recall  |
> |----------------|----------|--------|-----------|---------|
> | LLAMA2         | 0.2620   | 0.4317 | 0.4393    | 0.4314  |
> | LLAMA3         | 0.2715   | 0.4443 | 0.4528    | 0.4452  |
> | KULLM3         | 0.2600   | 0.4268 | 0.4373    | 0.4274  |
> | ExaOne         | 0.2505   | 0.4171 | 0.4297    | 0.4189  |
> | Mistral NEMO   | 0.2772   | 0.4491 | 0.4591    | 0.4476  |
>
> The tables present the results of the hallucination detection evaluation performed on the KHALU benchmark using translations of the Korean textual documents and statements into English and Malay. The results indicate that LLMs continue to struggle with hallucination detection, even when K-HALU is evaluated in different languages. The Korean version of K-HALU achieves approximately 2.7% higher performance compared to the translated versions, highlighting the influence of language-specific embedded knowledge on hallucination detection performance.
>
> **Q2. The lack of clarity on annotator consensus and their linguistic expertise raises concerns.**
>
> **R2.** We appreciate the reviewer's comment and have clarified the details of the human annotator consensus process (In 3.3 Quality Control; Ensemble Verification).
>
> **We have added the following content after lines 290-300**:
>
> *We employed three human annotators, all native Korean speakers, and graduates of four-year universities located in Seoul, Republic of Korea. The annotators reviewed 158 test examples flagged by at least one model as containing low-quality statements. They performed binary classification to determine whether revisions were necessary. A statement was considered for direct revision if two or more annotators agreed on the need for revision. As a result of the human evaluation, 137 test examples flagged as low-quality by the LLM meta-evaluators were found to reflect instances where the models either misinterpreted instructions or hallucinated during the evaluation process. For 21 test examples, at least two human annotators agreed that revisions were necessary. These problematic statements were subsequently revised by one of the authors, a native Korean speaker, and Ph.D. candidate. Krippendorff’s alpha for inter-annotator reliability was 0.923 among the three meta-evaluators and 0.828 among the three human annotators, indicating high and moderate inter-annotator agreement, respectively (Krippendorff, 2011).*

---

> ### Author Response · Authors · 2024-11-21
> **Response to Reviewer cEAG - Part 2**
>
> **Q3. The evaluation measure is a bit simple to some extent.**
>
> **R3.** We agree with the reviewer's suggestion. To increase the diversity of evaluation methods, **we have added the following two evaluation approaches in 4.1 Setup and 4.2 Results; Exact Match and LLM-as-a-Judge**:
>
> **Exact Match**: To enable LLMs to directly provide judgments, we modify the task to have the models generate binary outputs ("1" or "0") indicating the validity of each choice and evaluated them using exact match. We incorporate a post-processing step to minimize errors caused by unnecessary special characters or slight variations in format during the generation process. For example: [1, 1, 0, 0, 0].
>
> **LLM-as-a-Judge**: In this approach, LLMs are instructed to directly generate faithful or hallucinated statements based on the given task. The outputs are often descriptive, introducing a higher potential for errors when using exact match or overlap-based evaluations. To address this, we employ the GPT-4o model as an evaluator to classify the validity of the generated statements as binary values (1 for valid, 0 for invalid).
>
> **Exact Match Performance of LLMs**
>
> | Model          | 0-shot | 3-shot  | 3-shot + CoT |
> |----------------|-----------|---------|--------------|
> | LLAMA2         | 0.0106    | 0.0051  | 0.006        |
> | LLAMA3         | 0.1203    | 0.2396  | 0.176        |
> | KULLM3         | 0.0194    | 0.0083  | 0.007        |
> | ExaOne         | 0.0484    | 0.1885  | 0.1839       |
> | Mistral NEMO   | 0.124     | 0.2668  | 0.277        |
>
> When using exact match to evaluate performance, it becomes more challenging for models to achieve high accuracy compared to multiple-choice accuracy. This is due to the involvement of instruction-following abilities, which highlights more pronounced performance differences among models. Similar to evaluations based on multiple-choice and log probabilities, applying shots and CoT reasoning shows partial performance improvement. However, consistent performance enhancement is not observed across all settings.
>
> **LLM-as-a-Judge Performance of LLMs**
> | Model          | 0-shot | 3-shot  | 3-shot + CoT |
> |----------------|-----------|---------|--------------|
> | LLAMA2         | 0.0101    | 0.0115  | 0.0189       |
> | LLAMA3         | 0.0484    | 0.0203  | 0.0212       |
> | KULLM3         | 0.1134    | 0.1005  | 0.0797       |
> | ExaOne         | 0.0922    | 0.1276  | 0.1023       |
> | Mistral NEMO   | 0.0995    | 0.1138  | 0.1106       |
>
> To address potential distortions caused by descriptive outputs in exact match evaluations, we used GPT-4o to validate the generated results. However, even in this evaluation setup, baseline open-source LLMs consistently recorded very low performance. Interestingly, KULLM3 performed better in this evaluation compared to other methods. Additionally, ExaOne and Mistral NEMO demonstrated generally higher performance across log probability, exact match, and LLM-as-a-Judge evaluations.
>
> These results highlight the challenges open-source LLMs face in directly solving K-HALU’s hallucination detection tasks through generative outputs or CoT prompting. This implies the need for further research to improve model capabilities in hallucination detection tasks effectively.
>
> **Q4. More discussion on CoT can be helpful.**
>
> **R4.** The CoT prompts are designed to explain which part of the input document or statement contains errors, leading to the determination of hallucination. The prompts include three examples illustrating the reasoning process.
>
> For all three evaluation methods, **we have included the CoT prompts in Appendix G**, along with supplementary explanations. Below is an example of a CoT prompt. For brevity, considering the length constraints of OpenReview, the example has been reduced from three to one and includes a condensed version of the document content:
>
> **[CoT Prompt Example of Exact Match]**
>
> *Document Date: January 11, 2021*
>
> *Document Content: The Korean Association for Public Administration (KAPA) ... and then attended the live broadcast of former President Roh Moo-hyun’s inauguration the next day.*
>
> *Identify Hallucinated Sentences in the Document*
>
> *Options:*
>
> *1. Professor Soon-Ae Park strives to expand women’s participation in the public sector.*
>
> *Analysis: The document indirectly suggests efforts related to women but doesn’t explicitly state this.*
>
> *Conclusion: Not hallucinated → 0*
>
> ...
>
> *4. Professor Soon-Ae Park said she aims to change the administrative field, burdened by the heavy armor of the 3G era.*
>
> *Analysis: The document refers to the "2G era," not the "3G era."*
>
> *Conclusion: Incorrect information → 1*
>
> *5. Professor Soon-Ae Park believes civil servants prefer maintaining regulations over abolishing them.*
>
> *Analysis: While this is indirectly implied through comparisons with the U.S., it can be inferred as true.*
>
> *Conclusion: Factually supported → 0*
>
> *Answer: [0,0,1,1,0]*

---

> ### Author Response · Authors · 2024-11-21
> **Response to Reviewer cEAG - Part 3**
>
> **Q5. Precision, recall or F1 can be more comprehensive.**
>
> **R5.** Based on the reviewer's advice, **we have included evaluations of performance using precision, recall, and F1 scores in Appendix B**. As demonstrated earlier with the tables for the English and Malay experiments, we also measure F1, precision, and recall for the K-HALU Korean version. The results are shown in the table below:
>
> **K-HALU (Korean ver.)**
> | Model          | Accuracy | F1    | Precision | Recall  |
> |----------------|----------|-------|-----------|---------|
> | LLAMA2         | 0.2475   | 0.3939| 0.4152    | 0.3979  |
> | LLAMA3         | 0.2880   | 0.4332| 0.4503    | 0.4356  |
> | KULLM3         | 0.2862   | 0.4156| 0.4323    | 0.4174  |
> | ExaOne         | 0.3295   | 0.477 | 0.5117    | 0.476   |
> | Mistral NEMO   | 0.3014   | 0.4393| 0.453     | 0.4417  |
> | GPT-3.5        | 0.4046   | 0.7237| 0.6476    | 0.826   |
> | GPT-4          | 0.7346   | 0.8538| 0.8211    | 0.8899  |
> | GPT-4o         | 0.7866   | 0.8732| 0.8552    | 0.8924  |
>
> The Table presents the F1 score, precision, and recall metrics for evaluating model performance. These metrics are calculated using a macro-average approach, which considers the multiple correct answer distribution (6:3:1) and the uniform label distribution across the dataset. Accuracy requires selecting all correct answers to receive a score, meaning partially correct responses are treated as entirely incorrect. In contrast, the F1 score accommodates partial correctness, resulting in improved overall performance for the models. Open-source LLMs, such as LLAMA, KULLM3, and ExaOne, tend to predict only a subset of the correct answers, prioritizing those with higher log probabilities. This behavior leads to higher precision compared to recall, as these models adopt a more conservative strategy in their predictions. On the other hand, closed API LLMs, including GPT-3.5, GPT-4, and GPT-4 omni, generate answers more flexibly, often producing a greater number of options than the predefined correct answers. This results in higher recall than precision, as these models aim to capture all possible correct answers but occasionally overpredict.
>
> We sincerely thank Reviewer cEAG for the thoughtful review and valuable feedback. We hope our detailed responses have effectively addressed your concerns.

---

> > ### Comment · Reviewer_cEAG · 2024-11-24
> > **feedback on author response**
> >
> > The reviewer thanks the authors for making the responses and providing additional results. Some of the concerns about the data annotation and experimental results have been addressed, thus the reviewer raised the score accordingly.

---

> > > ### Author Response · Authors · 2024-11-24
> > > **Thanks to the Reviewer cEAG**
> > >
> > > We sincerely thank the reviewer for their thoughtful feedback and for taking the time to re-evaluate our work. We are pleased that our responses and additional results helped to address some of your concerns regarding data annotation and experimental results.

---

### Author Response · Authors · 2024-12-04
**Reflections and Summary of Discussions**

Dear Area Chair and Reviewers,

Thank you for your support and thoughtful feedback throughout the discussion phase. We truly appreciate the time and effort you have invested in reviewing and providing valuable insights on our submission. The summary of revisions to our manuscript is as follows:

1. **Expansion and justification of evaluation methods**
   - Added F1 score for partial credit scoring.
   - Incorporated exact match and LLM-as-a-Judge evaluation methods.
   - Provided a detailed explanation for adopting the multiple choice and multiple answer formats.

2. **Multilingual extension**
   - Expanded the dataset to include high-resource language (English) and low-resource language (Malay).

3. **CoT prompt details**
   - Added examples of CoT prompts and strengthened explanations.

4. **Human annotation details**
   - Included the annotators' background and inter-annotator agreement metrics.

5. **Dataset license**
   - Clearly specified the dataset license.

6. **Content reorganization**
   - Adjusted the placement of content between the main text and the appendix.

7. **Terminology refinements**
   - Corrected terminology and added explanations for previously omitted details.

We are sincerely grateful for the constructive discussions and the opportunity to refine our work.

Best regards,
Authors

---

### Meta-Review · Area_Chair_qUcU · 2024-12-19

**Metareview:**

This paper introduces a multiple-answer Korean benchmark K-HALU to evaluate LLMs’ hallucination detection in Korean. It consists of 2,170 test samples, each including a textual document, a publish date, an instruction, and statements. Several multilingual LLMs are evaluated on the benchmark and the results show that open-source LLMs still have difficulty with hallucination detection in Korean knowledge. The proposed benchmark may be a valuable resource for future research, especially in the Korean NLP community. The paper is generally easy to follow. The reviewers raised major concerns about the presentation, such as lack of clarity regarding the human-annotated statements, lack of comparisons and discussion with the English benchmarks, the unclear rationale of the format of the benchmark and the task setting, etc. Most of the concerns have been addressed in the authors' rebuttal. To me, this work has limited novelty, as the dataset construction process doesn't involve any innovative technique and there are already quite a few similar benchmarks in other languages. It would be more interesting if the benchmark covers multiple Asian languages and other task formats. In summary, though the reviewers' scores are positive, I still have mixed feelings about this paper and it is a borderline one.

**Additional Comments On Reviewer Discussion:**

The reviewers increased their scores during the rebuttal period and they were finally positive about this paper.

---

### Decision · Program_Chairs · 2025-01-22

Accept (Poster)